# UGM2N: An Unsupervised and Generalizable Mesh Movement Network via M-Uniform Loss

**Zhichao Wang**[1,2]    **Xinhai Chen**[1,2*]  **Qinglin Wang**[1,2] **Xiang Gao**[1,2]
**Qingyang Zhang**[1,2]    **Menghan Jia**[1,2]    **Xiang Zhang**[1,2]    **Jie Liu**[1,2]
[1]Laboratory of Digitizing Software for Frontier Equipment,
National University of Defense Technology, Changsha, China,
[2]National Key Laboratory of Parallel and Distributed Computing,
National University of Defense Technology, Changsha, China,
wangzhichao@nudt.edu.cn, chenxinhai16@nudt.edu.cn

## Abstract

Partial differential equations (PDEs) form the mathematical foundation for modeling physical systems in science and engineering, where numerical solutions demand rigorous accuracy-efficiency tradeoffs. Mesh movement techniques address this challenge by dynamically relocating mesh nodes to rapidly-varying regions, enhancing both simulation accuracy and computational efficiency. However, traditional approaches suffer from high computational complexity and geometric inflexibility, limiting their applicability, and existing supervised learning-based approaches face challenges in zero-shot generalization across diverse PDEs and mesh topologies. In this paper, we present an **U**nsupervised and **G**eneralizable **M**esh **M**ovement **N**etwork (UGM2N). We first introduce unsupervised mesh adaptation through localized geometric feature learning, eliminating the dependency on pre-adapted meshes. We then develop a physics-constrained loss function, M-Uniform loss, that enforces mesh equidistribution at the nodal level. Experimental results demonstrate that the proposed network exhibits equation-agnostic generalization and geometric independence in efficient mesh adaptation. It demonstrates consistent superiority over existing methods, including robust performance across diverse PDEs and mesh geometries, scalability to multi-scale resolutions and guaranteed error reduction without mesh tangling.

## 1 Introduction

Solving partial differential equations (PDEs) is fundamental for modeling physical phenomena, spanning fluid dynamics, heat transfer, quantum mechanics, and financial markets [1]. Modern PDE solving critically relies on meshes, which serve as the foundational discretization framework for numerical methods [2, 3]. The accuracy and computational cost of PDE solutions are significantly affected by mesh resolution: high-resolution meshes resolve complex physics at high computational expense, whereas coarse meshes improve efficiency but risk missing critical features. As problem complexity grows, geometric details demand exponentially finer resolution, while multi-physics interactions require dynamic adaptation—pushing memory, parallel efficiency, and solver convergence to their limits [4]. To alleviate this issue, mesh adaptation methods—particularly mesh refinement method ($h$-adaptation method) and mesh movement method ($r$-adaptation method) —dynamically optimize computational resources, systematically overcoming traditional bottlenecks through intelligent spatial discretization control [5–7].

---

[*]Corresponding Author

39th Conference on Neural Information Processing Systems (NeurIPS 2025).

Mesh refinement method dynamically adjusts resolution via local element subdivision/coarsening, altering node counts while retaining fixed positions. In contrast, mesh movement method preserves node counts but relocates them strategically to high-resolution regions, guided by error estimators or gradients [8]. While mesh refinement method handles discontinuities via topological changes, mesh movement method suits smooth domains, avoiding remeshing overhead. However, the traditional Monge-Ampère (MA)-based methods suffer from high computational costs due to (1) repeated mesh-motion PDE solves (e.g., solving auxiliary equations) and (2) mesh-quality checks to prevent inversion. In extreme cases, adaptive operations can exceed the PDE-solving cost itself, making the enhancement of mesh adaptation efficiency an enduring open problem.

To improve the efficiency of mesh movement method, pioneering works employ supervised learning, training models on meshes adapted via traditional MA-based methods. Song et al. [9] propose a mesh adaptation framework trained via MSE loss between initial and adapted mesh nodes, and Zhang et al. [10] introduce a zero-shot adaptive model trained with a combined loss of volume preservation and Chamfer distance between initial and adapted mesh nodes. However, such supervised methods exhibit limited generalization: M2N requires PDE- and geometry-specific retraining, risking mesh tangling under extreme deformations, while UM2N's zero-shot performance may degrade for unseen domains or PDEs.

In this paper, we propose UGM2N, an unsupervised and generalizable mesh movement network. Inspired by vision Transformers [11], we introduce node patches, locally normalized nodes with first-order neighbors, as model inputs. Unlike M2N/UM2N's whole-mesh processing, our method parallelly and independently computes adapted positions for each patch, simplifying the learning objective and enabling scale-invariant mesh adaptation. Leveraging the node patch representation, we formulate an M-Uniform loss function that mathematically encodes local equidistribution properties, the core objective of mesh movement methods. Minimizing this patch-wise loss can produce approximately M-Uniform meshes while effectively matching MA-based adaptation objectives—all achieved through an efficient unsupervised framework. By learning adaptation dynamics directly, our model achieves equation-agnostic generalization while maintaining mesh-geometric independence.

Our main contributions are summarized as follows:

- We present an unsupervised mesh movement network, eliminating the need for pre-adapted meshes by learning solely on initial meshes and flow fields. Our novel node-patch representation processes localized neighborhoods rather than full meshes, enabling efficient training and inherent generalization.

- We derive a theoretically grounded M-Uniform loss function that enforces local mesh equidistribution at the node-patch level, which aligns with MA-based optimization objectives through a fully data-driven approach with native equation-agnostic generalization across arbitrary mesh geometry.

- We present extensive numerical validation showing exceptional generalizability and robustness across various PDE types (both steady-state and time-dependent), accommodating different boundary conditions or initial conditions, and mesh geometries with varying shapes or resolutions.

## 2 Related Work

**Machine learning for mesh generation and optimization.** The automation and intelligence of mesh generation are among the key challenges in CFD 2030 [12], driving significant research efforts toward intelligent mesh generation and optimization. Zhang et al. [13, 14] proposed the MeshingNet and MeshingNet3D models to generate high-quality tetrahedral meshes, demonstrated in linear elasticity problems on complex 3D geometries. Chen et al. [15] introduced the MGNet model, which employs physics-informed neural networks [16] to achieve structured mesh generation. For mesh optimization, Guo et al. [17], Wang et al. [18, 19] developed intelligent mesh optimization agents based on supervised learning, unsupervised learning, and reinforcement learning (RL), achieving a balance between optimization efficiency and quality.

**Machine learning for mesh adaptation.** Unlike static mesh optimization methods, mesh adaptation dynamically modifies the computational mesh during simulation to enhance resolution in critical regions (e.g., shock waves, boundary layers, or vortex-dominated flows). These techniques are

guided by error estimation schemes or feature-based criteria, ensuring computational efficiency while preserving accuracy. Advanced implementations leverage machine learning to predict optimal adaptation strategies, enabling high-fidelity simulations for complex, evolving flows.

For mesh refinement method, Foucart et al. [20] pioneered RL for adaptive mesh refinement, formulating it as a POMDP and training policy networks directly from simulations. Dzanic et al. [21] developed DynAMO, using multi-agent RL to predict future solution states for anticipatory refinement. Kim et al. [22] introduced GMR-Net, leveraging graph CNNs to predict optimal mesh densities without costly error estimation. Beyond these foundational works, research in intelligent $h$-adaptive mesh refinement remains highly active, with additional advancements documented in [23–29].

For mesh movement method, Omella and Pardo [30] proposed a neural network-enhanced boundary node optimization method, which is specifically designed for tensor product meshes. Song et al. [9] introduced M2N, a framework combining neural splines with Graph Attention Network (GAT) [31], enabling end-to-end mesh movement with 3–4 order-of-magnitude speedups. Hu et al. [32] introduced a neural mesh adapter trained via the MA equation physical loss to dynamically adjust mesh nodes, and develops a moving mesh neural PDE solver that improves modeling accuracy for dynamic systems. Rowbottom et al. [33] proposed a graph neural diffusion method that directly minimizes finite element error to achieve efficient mesh adaptation, and proved that its model architecture can effectively avoid mesh entanglement. For specialized applications, methods such as Flow2Mesh and Para2Mesh have demonstrated the efficacy of learning-based adaptation in aerodynamic simulations [34, 35]. Recent work extended these advances with UM2N [10], a universal graph-transformer architecture attempts to achieve zero-shot adaptation across diverse PDEs and geometries. Most of the aforementioned works rely on supervised learning, where models are trained to align their outputs with pre-adapted meshes, resulting in a lack of physical information. Additionally, they often require retraining for different PDEs or mesh geometries, limiting their generalizability. This paper adopts an unsupervised learning approach to achieve equation-agnostic generalization across arbitrary mesh geometries.

## 3 Method

### 3.1 Problem statement and preliminaries

Given an initial mesh $\mathcal{M}$ (e.g., a uniform mesh) and associated flow field variables (such as velocity $\mathbf{u}$ or pressure $p$), the mesh movement method optimizes the node positions to generate an adapted mesh satisfying predefined resolution criteria. The mesh movement process can be analyzed from different perspectives, such as coordinate mapping between uniform and adapted meshes, uniform mesh construction in metric space, and so on [6]. The MA-based method adopts the former approach, solving the MA equation with boundary conditions to obtain the coordinate mapping (for a detailed introduction, refer to App. A). In contrast, this study employs the latter perspective, enforcing uniform distribution in metric space without explicit coordinate transformations between computational and physical domains.

From the latter perspective, given a physical domain $\Omega \subset \mathbb{R}^d$ (where $d \geq 1$), the goal of mesh movement is to construct uniform meshes in some metric space, which is defined by a matrix-valued monitor function $M = M(\mathbf{x})$, where $\mathbf{x} \in \Omega$. A mesh is said to satisfy the *mesh equidistribution condition* if it is uniformly distributed in this metric space, which can be mathematically expressed as:

$$\int_K m(\mathbf{x})d\mathbf{x} = \frac{\sigma}{N_e}, \forall K \in \mathcal{M}, \tag{1}$$

where $\sigma = \int_\Omega m(\mathbf{x})\, d\mathbf{x}$, $m(\mathbf{x}) = \sqrt{\det(M(\mathbf{x}))}$ is the mesh density function, $K$ is the element of $\mathcal{M}$, and $N_e$ is the number of elements in the mesh $\mathcal{M}$. This condition constrains the size of mesh elements—when $m(\mathbf{x})$ is large, the element volume should be small, and vice versa. Additionally, the M-Uniform mesh condition also requires that the mesh elements should be equilateral in the metric space. In this work, we primarily focus on modifying the mesh density while disregarding the equilateral alignment of mesh elements.

Compared to MA-based coordinate transformations, this approach for constructing uniform meshes in the metric-space offers a more discretization-friendly framework, particularly well-suited for local loss function modeling (see Section 3.3).

## 3.2 Network overview

The proposed UGM2N is illustrated in Fig. 1. The model takes the initial mesh with solution as input, and node patches are constructed from all mesh nodes, with input features generated using flow field variables. The coordinates of mesh nodes within each patch are normalized to $[0, 1] \times [0, 1]$ via 0-1 normalization, then encoded through node and edge encoders to obtain embeddings. These embeddings are processed by multiple deform blocks and a node decoder, producing adapted node coordinates for each patch, which are denormalized to restore the original mesh. The flow field features of the updated mesh are obtained through Delaunay Triangulation-based interpolation on the original mesh, and the adapted mesh serves as the initial input for the next iteration, with the process repeating for a maximum of $E$ epochs.

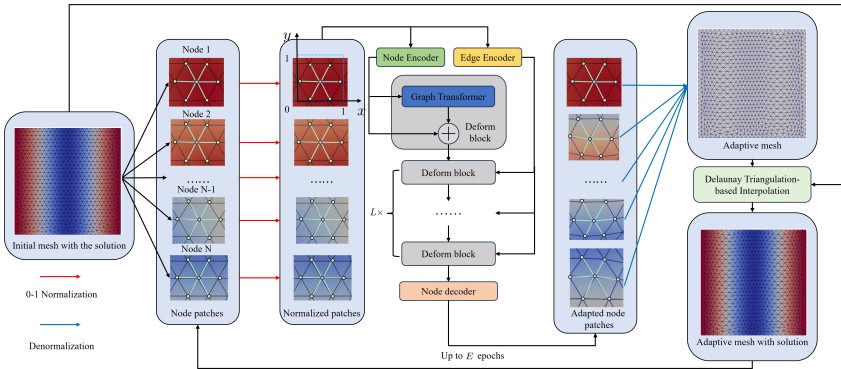

Figure 1: The proposed mesh movement network.

**Node patches.** Inspired by vision Transformers [11], the proposed model processes individual mesh node patches, unlike M2N or UM2N, which take the entire mesh as input. This patch-based approach significantly improves local feature representation while maintaining computational efficiency and scalability, mirroring the local optimization principles employed in mesh smoothing techniques.

In each adaptive epoch, the input consists of an initial mesh with a flow field solution, denoted as $\mathcal{M} = \{\mathcal{V}, \mathbf{U}, \mathcal{E}\}$, where $\mathcal{V} = \{\mathbf{x}_1, \ldots, \mathbf{x}_N\}$ represents node coordinates, $\mathcal{E}$ denotes connectivity, and $\mathbf{U} = \{\mathbf{u}_1, \ldots, \mathbf{u}_N\}$ contains flow variables on the nodes. A node patch $\mathcal{P}_i = \{\mathbf{X}_i, \mathcal{E}_i\}$ is defined as the node itself, its first-order neighbors, and their connections (excluding inter-neighbor connections). Patch normalization scales nodes to a unit square, reducing learning difficulty and accommodating varying mesh sizes. It is worth noting that normalization does not introduce additional computational overhead, as it can be efficiently implemented using vectorized operations.

The flow field features are incorporated into the patch features using a mesh density function derived from the Hessian matrix:

$$M(\mathbf{x}_i) = \left(1 + \alpha \frac{\|\mathbf{H}(u_i)\|}{\max_j \|\mathbf{H}(u_j)\|}\right) I, \tag{2}$$

$$m(\mathbf{x}_i) = 1 + \alpha \frac{\|\mathbf{H}(u_i)\|}{\max_j \|\mathbf{H}(u_j)\|}, \tag{3}$$

where $\alpha$ is a constant, $u_i = \|\mathbf{u}_i\|_2$, $I$ is the identity matrix, and $\|\mathbf{H}(u_i)\|$ is the Frobenius norm of the Hessian. The scalar $m(\mathbf{x}_i)$ is concatenated with node coordinates (yielding a 3D input for 2D meshes) as the model's input.

**Mesh coordinate computation based on Graph Transformer model.** The adapted node coordinates are computed using a lightweight model. Node and edge features (central-to-neighbor coordinate vectors) are first encoded via dedicated MLP encoders, followed by $L$ deform blocks for graph feature extraction. We employ a residual-connected Graph Transformer [36] in each block, proving effective despite its simplicity. The adapted patch coordinates are then computed through the node MLP decoder, and the *centering* mesh node within each patch (the pink node in Fig. 1) is denormalized to the original mesh space through vectorized operations. For boundary nodes, we ignore them and do not perform adaptation, since the output coordinates are unlikely to lie precisely on the boundary.

**Iterative mesh adaptation with dynamic termination.** Inspired by iterative mesh smoothing, multi-epoch mesh adaptation is employed to progressively refine node distribution during inference[2], the Hessian norm values are updated between epochs via Delaunay triangulation-based interpolation from original to adapted nodes, providing initialization for subsequent adaptations. While this process could theoretically continue indefinitely, convergence is not guaranteed and mesh validity may degrade (see App. B.2 for discussions). A fixed epoch count would limit optimization capability; instead, we propose a metric-based adaptive strategy that dynamically determines termination based on optimization progress. This approach will be detailed following the presentation of our unsupervised loss function.

### 3.3  M-Uniform loss

Unlike existing methods, which adopted supervised loss functions to align predicted meshes with reference meshes, our approach addresses two key challenges in practical applications: (1) the frequent unavailability of high-quality reference meshes, especially for multi-physics or geometrically complex problems, and (2) the poor zero-shot generalization to novel PDE types beyond the training distribution. These limitations motivate our development of an unsupervised adaptation method.

As introduced in the Section 3.1, enforcing the *mesh equidistribution condition* offers a novel approach. Eq. 1 requires that the integral of $m(\mathbf{x})$ over any mesh element $K$ be constant. However, since $m(\mathbf{x})$ is only known at mesh nodes, exact integration is infeasible. We thus relax the *strict M-Uniform condition* to an *approximate M-uniform condition*:

$$\int_K m_K d\mathbf{x} = m_K |K| = \frac{\sigma_h}{N_e}, \forall K \in \mathcal{M}, \tag{4}$$

$$m_K = \sqrt{\det(M_K)}, M_K = \frac{1}{|K|} \int_K M(\mathbf{x})d\mathbf{x}, \tag{5}$$

where $|K|$ represents the volume of the mesh element $K$ and $\sigma_h$ is a constant. Here, $M_K$ is approximated via nodal averages: for a triangular element $K$ with nodes $K_1, K_2, K_3$, $M_K = \frac{1}{3}\sum_{j=1}^3 M(\mathbf{x}'_{K_j})$ (note that $M(\mathbf{x}')$ require interpolation to obtain), where $\mathbf{x}'_i$ is the adapted position of node $i$ output by the model. Together with Eq. 2 and 3, we can obtain $m_K = \frac{1}{3}\sum_{j=1}^3 m(\mathbf{x}'_{K_j})$. Building upon these foundations, we can define a metric function for element $K$:

$$\mathcal{L}_K = m_K |K|. \tag{6}$$

Let $K_l^i$ be the mesh element $l$ in the patch of mesh node $i$. Rewriting Eq. 4 in terms of the local mesh node $i$, we require that $\mathcal{L}_{K_l^i}$ be as uniform as possible around mesh node $i$. We measure the variation in $\mathcal{L}_{K_l^i}$ among different mesh elements using a variance-based loss function:

$$\mathcal{L}_{\text{var}}(\mathcal{P}_i) = \frac{1}{N_i} \sum_{l=1}^{N_i} \left( \mathcal{L}_{K_l^i} - \overline{\mathcal{L}_{K_l^i}} \right)^2, \tag{7}$$

where $N_i$ is the number of mesh elements in the patch $\mathcal{P}_i$, and $\overline{\mathcal{L}_{K_l^i}} = \frac{1}{N_i}\sum_{l=1}^{N_i} \mathcal{L}_{K_l^i}$. Then, the proposed M-Uniform loss function can be written as:

$$\mathcal{L}_M(\theta) = \lambda \mathbb{E}_{i \in \{1,...,N\}} \mathcal{L}_{\text{var}}(\mathcal{P}_i), \tag{8}$$

where $\theta$ is the model parameters, and $\lambda = 100$ is a scaling constant. This approach shares conceptual similarities with PINNs, where local constraints—residual conditions in PINNs and the M-Uniform condition here—guide the learning process. By enforcing mesh equidistribution condition at the node level, the model can adapt mesh node positions without supervised data, ensuring generalization to arbitrary adaptive scenarios. Crucially, unlike existing adaptive methods (e.g., M2N or UM2N), training does not require the full mesh as input. Instead, it can be trained on individual mesh nodes. For example, during mini-batch training, we can sample a subset of mesh nodes from the mesh and achieve efficient training through graph batching[3]. Moreover, this approach further reduces the

---

[2]Multi-epoch mesh adaptation is disabled during training to simplify the training process.

[3]Notably, **during training**, we do not use the model outputs to update the mesh nodes—in other words, there is no dynamic mesh update.

required amount of data, as the number of data samples is proportional to the number of mesh nodes rather than the number of meshes.

During iterative mesh adaptation, we compute the global uniformity metric $\mathcal{L}_{\text{var}}(\mathcal{M}')$ over the entire adapted mesh $\mathcal{M}'$ after each epoch to assess equidistribution compliance:

$$\mathcal{L}_{\text{var}}(\mathcal{M}') = \frac{1}{N_e} \sum_{l=1}^{N_e} \left( \mathcal{L}_{K_l} - \overline{\mathcal{L}_{K_l}} \right)^2, \tag{9}$$

where $N_e$ is the number of mesh elements in $\mathcal{M}'$, and $\overline{\mathcal{L}_{K_l}} = \frac{1}{N_e} \sum_{l=1}^{N_e} \mathcal{L}_{K_l}$. The model stops the iterative mesh adaptation when $\mathcal{L}_{\text{var}}(\mathcal{M}')$ no longer decreases. During inference, we set a fixed upper limit for the number of iterations—specifically, we set the maximum number of mesh adaptation epochs as 10. The full adaptation algorithm is detailed in App. B.1, and the theoretical analysis of the effectiveness of the loss function to optimize mesh distribution is provided in App. B.2.

## 4 Experiment

### 4.1 Experiment setups

Different numerical simulations involve diverse flow-field and mesh geometries characteristics. An effective mesh movement method must account for variations in both the flow field and the underlying mesh geometry. Our experiments show that the proposed method achieves robust, optimal performance across both scenarios—whether applied to different flow fields (diverse PDEs with varying solutions) or entirely distinct mesh geometries.

**Model training.** Following UM2N's protocol, we trained UGM2N on a mesh with only **four flow fields** and evaluated its zero-shot generalization performance on unseen flow fields or meshes. As depicted in Fig. 12 (App. C.1), random perturbations were applied to mesh node positions to enhance data diversity, resulting in a training set comprising 10,440 mesh nodes. The model was optimized using Nadam [37] with an initial learning rate of 1e-4, with all experiments conducted on an NVIDIA RTX TITAN GPU. See App. D for more training details.

**Baselines and metrics.** We performed a comparative analysis against the MA method [38], M2N, and UM2N, using UM2N's pre-trained weights obtained from its GitHub repository [39]. Performance was evaluated using two key metrics: (1) error reduction (ER), which measures the relative improvement in PDE solution accuracy compared to the initial coarse mesh (with the high-resolution solution serving as the reference), and (2) tangling ratio (TR), defined as the fraction of invalid elements in the adapted mesh. Additional case-specific metrics will be presented in the corresponding experimental sections. Detailed mathematical definitions of all metrics are provided in App. E.1.

### 4.2 Performances across different flow field solutions

To assess the model's generalization ability across diverse flow fields, we conducted experiments using Burgers' equation with varying initial conditions, as well as Poisson and Helmholtz equations with different analytical solutions (refer to App. C.3 and C.4 for more detailed PDE configurations). All simulations were performed on a uniform triangular mesh spanning the domain $[0, 1] \times [0, 1]$, comprising 1,478 elements. For validation, a high-fidelity reference solution was computed on a significantly refined mesh with 23,250 elements, ensuring precise accuracy for benchmarking purposes.

The test results are summarized in Table 1, it can be observed that the our model demonstrates significant advantages in the vast majority of cases. In the seven test cases for the Poisson equation, our model achieved optimal performance (highest ER or lowest TR) in five cases, particularly excelling with complex functions. For example, for $\sum_{i,j} \exp[-\frac{(x-x_{\mu,i})^2}{0.25^2} - \frac{(y-y_{\mu,j})^2}{0.25^2}]$, ours achieved an ER of 9% , far surpassing the comparison methods. Additionally, ours achieved complete dominance in the Helmholtz equation, securing the four highest ER out of five test cases. Although slightly inferior to M2N in the Burgers equation, ours still significantly outperformed UM2N. Moreover, in all cases, our model did not produce any mesh tangling phenomenon (for the mesh tangling test on non-convex meshes, refer to App. E.2). These results validate the robustness and generalization capability of our

Table 1: Model performance of different flow fields

| PDEs | Variables | ER(%)↑ or TR(%)↓ | | | |
|---|---|---|---|---|---|
| | | MA [38] | M2N [9] | UM2N [40] | Ours |
| **Poisson** | $u_{\text{exact}}$ | | | | |
| | $1 + 8\pi^2 \cos(2\pi x)\cos(2\pi y)$ | **15.40** | 0.92 | 6.74 | 14.56 |
| | $\sum_{i,j} \exp\left[-\left(\frac{x-x_{\mu,i}}{0.25}\right)^2 - \left(\frac{y-y_{\mu,j}}{0.25}\right)^2\right]$ [4] | -8.64 | -30.20 | -5.59 | **9.00** |
| | $\sin(4\pi x)\sin(4\pi y)$ | 9.79 | -98.01 | -2.19 | **12.46** |
| | $1/\exp((x-0.5)^2 + (y-0.5)^2)$ | -28.22 | 1.15 | -2.98 | **1.70** |
| | $\sin(2\pi x + 2\pi y)$ | 11.69 | -34.03 | -9.07 | **9.07** |
| | $\cos(\pi x)\exp\left(-((x-0.5)^2 + (y-0.5)^2)\right)$ | -8.94 | -41.89 | **5.15** | 4.90 |
| | $\cos\left(\sqrt{(x-0.5)^2 + (y-0.5)^2}\right) \times$ $\exp\left(-((x-0.5)^2 + (y-0.5)^2)\right)$ | -25.23 | 1.62 | -3.53 | **2.56** |
| **Helmholtz** | $u_{\text{exact}}$ | | | | |
| | $\cos(2\pi y)$ | **15.60** | -11.16 | 10.86 | 14.11 |
| | $\cos(2\pi x)$ | 10.29 | -37.24 | 6.80 | **13.15** |
| | $\cos(2\pi y)\cos(2\pi x)$ | 13.48 | -24.33 | 5.63 | **15.03** |
| | $\cos(2\pi y)\cos(4\pi x)$ | 10.87 | -351.63 | -2.61 | **14.09** |
| | $\cos(4\pi y)\cos(2\pi x)$ | 13.50 | -250 | 3.43 | **16.98** |
| **Burgers** | $\mathbf{u}_{\text{ic}}$ | | | | |
| | $\left[\sin\left(-20\left(x-0.5\right)^2\right), \cos\left(-20\left(y-0.5\right)^2\right)\right]^T$ | 26.81 | **44.82** | 0.46 | 32.22 |
| | $\left[\exp\left(-\left((x-0.5)^2 + (y-0.5)^2\right) \times 100\right), 0\right]^T$ | 51.12 | 29.93 | 22.76 | 30.19 |

method in mesh adaptation across different PDEs, achieving state-of-the-art performance compared to existing approaches.

The mesh adaptation results for the Helmholtz equation are shown in Fig. 2 (for results on the Poisson and Burgers equation, refer to App. E.3). Our method demonstrates superior visual alignment with the target flow field while simultaneously generating meshes with excellent quality compared to alternative approaches. It is noteworthy that, although M2N and UM2N can generate qualitatively adaptive meshes, they exhibit weaker compliance with the mesh equidistribution condition compared with our method (see App. E.5). Additionally, in the present work the boundary nodes are kept fixed during adaptation; experiments that allow constrained movement of boundary nodes are reported in App. E.4.

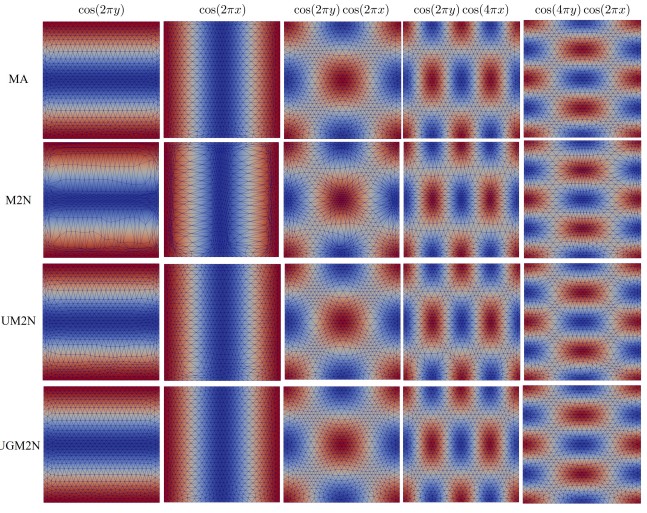

Figure 2: Mesh adaptation results for the Helmholtz equation with different solutions.

## 4.3 Performance across varying mesh geometries

**Varying mesh resolutions.** Practical simulations often involves meshes with varying element sizes, making it crucial for the model to handle meshes of different resolutions. We tested the model's performance on the Helmholtz equation with different mesh element sizes, where the solutions are the same as in Table 1. The coarse mesh element sizes were [0.05, 0.04, 0.03, 0.02], corresponding to mesh element counts of [944, 1478, 2744, 5824]. As shown in Fig. 3, our model achieved improved solution accuracy across all element sizes, whereas the M2N model failed to adapt the mesh at any resolution, and UM2N could only generalize on some of the element sizes. The results demonstrate

---

[4] $\mathbf{x}_\mu = [0.25, 0.25]^T, \mathbf{y}_\mu = [0.25, 0.25]^T$

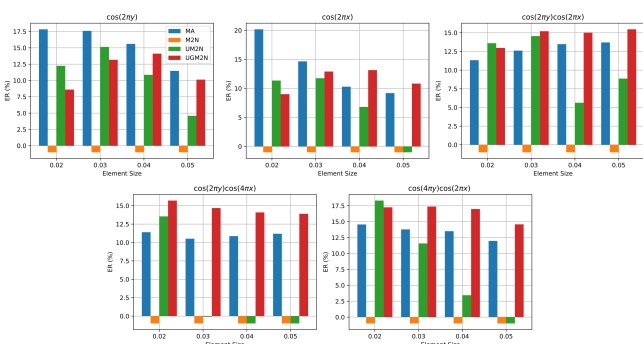

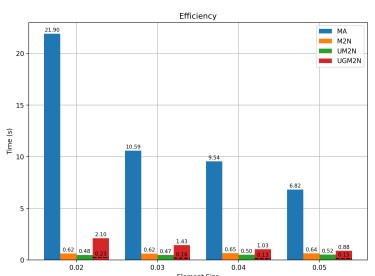

Figure 3: The ER for different mesh resolutions on the Helmholtz equation. For clarity, we clipped the minimum ER at -1%, even though for some methods (e.g., M2N), their adapted meshes significantly increased the solution error.

Figure 4: The performance of mesh movement methods under different mesh resolutions is presented. The dashed line in the figure indicates the time required for UGM2N to complete a single iteration.

that the loss function based on MSE between mesh nodes in M2N struggles to generalize to unseen meshes. Furthermore, UM2N's volume loss exhibits only limited generalizability.

Regarding computational efficiency, as illustrated in Fig. 4, we present the time required for mesh movement under varying element sizes. For each model configuration, we conducted ten repeated tests on different solutions of the Helmholtz equation and reported the average mesh movement time per trial. Compared with the MA method, the neural-based mesh movement method demonstrates significant advantages. For detailed efficiency and scalability analysis, see App. E.6.

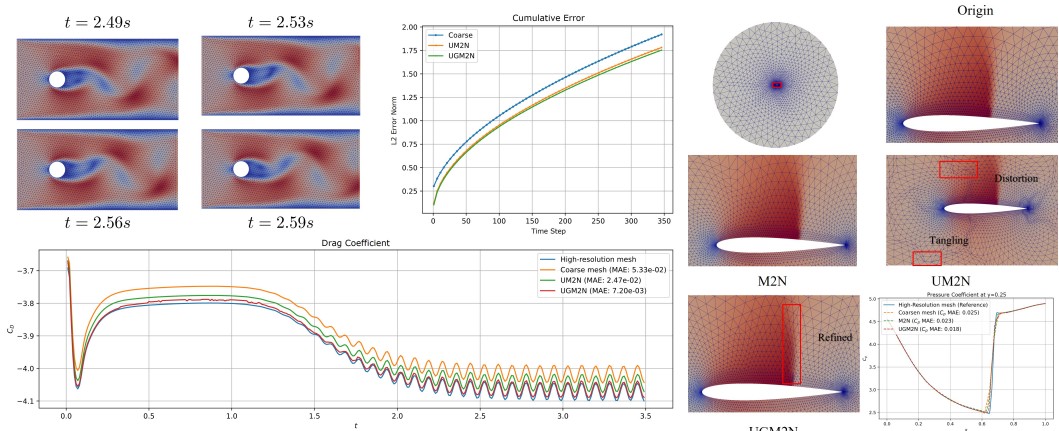

Figure 5: Results on the cylinder flow. We present the adaptive meshes at four time slices, with the results over the entire simulation period provided in the supplementary video materials.

Figure 6: Mesh adaptation on the subsonic flow case. MA method fails to converge in this case.

**Varying mesh shapes.** Another potential variable in the simulation is the shape of mesh. To evaluate the model's generalization capability under different mesh geometric configurations, we quantitatively conducted three distinct simulation cases: subsonic flow around a NACA0012 airfoil, cylinder flow, and the wave equation on a circular domain, with the experimental setups provided in App. C.3. The adaptive results of the airfoil mesh are shown in Fig. 6, which shows that our model effectively captures the shock wave location without introducing any invalid elements. Additionally, at the position $y = 0.25$, our model obtains the minimum mean absolute error (MAE [5]) in pressure coefficient result. For the cylinder flow case, neither MA nor M2N could produce valid meshes. As

---

[5]The average absolute error between the solution on the high-resolution mesh and the solution on the adaptive (coarse) mesh at each position (time step).

shown in Fig. 5, compared to UM2N, our model further reduces the prediction error of the drag coefficient and slightly decreasing the cumulative error during the solving process. The results of the wave equation are shown in Fig. 7. At any given time step, our proposed method consistently generates smooth, high-quality adaptive meshes. In contrast, other methods may produce distorted mesh elements and fail to reduce errors, although they can also generate adaptive meshes.

To further validate UGM2N's generalization to more complex mesh topologies, we demonstrated its capability on irregular and anisotropic meshes in App. E.7, where the regularity assumption (App. B.2) may not hold. Additionally, we benchmarked all methods on 1,000 random polygonal domains using Gaussian mixture flow fields as exact solutions in App. E.8. UGM2N achieves a Positive ER Ratio of 0.807 with a stable mean ER of 13.99% ± 21.05%, decisively outperforming MA (0.110) and UM2N (0.245), both of which frequently yield negative ER. These results demonstrate that UGM2N achieves substantial error reduction and robust adaptation even in challenging scenarios.

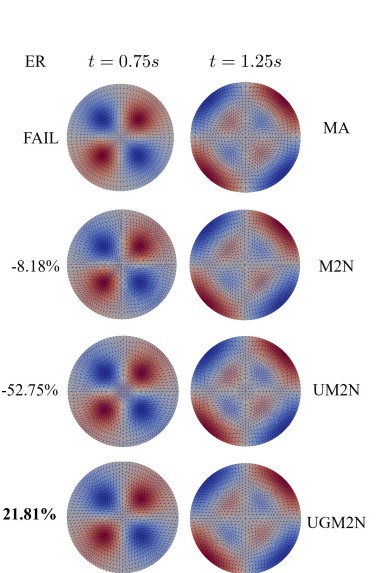

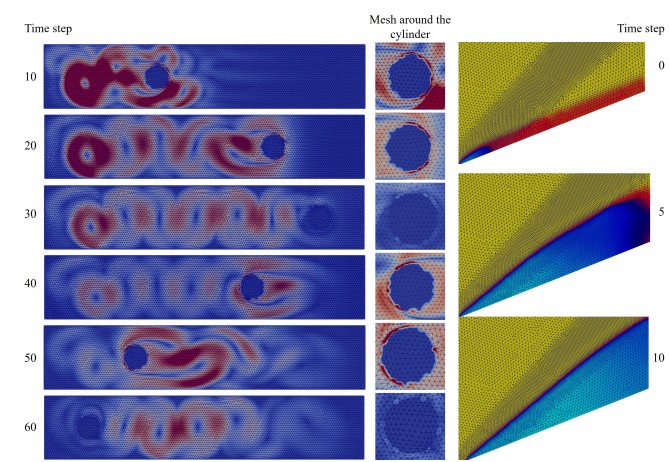

Figure 8: Mesh adaptation on moving cylinder and supersonic flow cases. In supersonic flow, UGM2N's adaptive mesh dynamically adjusts nodes to track shock waves; in the moving cylinder case, it precisely captures both the cylinder boundary and wake flow. The UM2N results for M2N are given in App. E.9. The results are also presented in the supplementary video.

Figure 7: Mesh adaptation results on the wave equation.

Additionally, three qualitative experiments—supersonic flow over a wedge, moving cylinder in a channel, and Tohoku tsunami with complex boundaries—are provided (first two in Fig. 8, third in supplementary materials), demonstrating effective adaptive meshes for realistic simulation scenarios.

### 4.4 Ablation study

**Loss function** To validate the effectiveness of the M-Uniform loss compared to the coordinate loss of M2N and the volume loss of UM2N, we trained models with the same architecture and the same training data but different loss functions. The supervised data was generated using the MA method. Table 2 presents the average ER on PDEs with different solutions of models trained with different loss functions on three types of equations (the equation configurations are the same as in Table 1). With only a small amount of training data, supervised learning methods using the entire mesh as input struggle to produce effective models. In contrast, our unsupervised training approach requires no adaptive meshes as supervised data, and the node patch-based training method enables effective model training even with limited data. App. C.2 analysis shows increasing training data (10,440 to 41,760 patches) improves ER by up to 43% for Poisson and Helmholtz cases, indicating richer datasets further enhance model optimization.

**Iterative mesh adaptation** To demonstrate the effectiveness of our iterative mesh adaptation approach, Fig. 9 shows the error reduction across optimization iterations for the Poisson equation in

Table 1. The results reveal that the error reduction exhibits a non-monotonic trend, initially increasing before decreasing as optimization progresses. Notably, our adaptive adaptation epochs (marked with a diamond) consistently stay within high ER regime, demonstrating the effectiveness of our method.

Table 2: The mesh adaptation performance of models trained with different loss functions

|  | ER (%) ↑ | | |
| Loss | Poisson | Helmholtz | Burgers |
| --- | --- | --- | --- |
| Coordinate loss | -8.19 | -4.46 | -9.17 |
| Volume loss | -8.27 | -0.52 | -1.46 |
| M-Uniform loss | **5.21** | **9.94** | **30.07** |

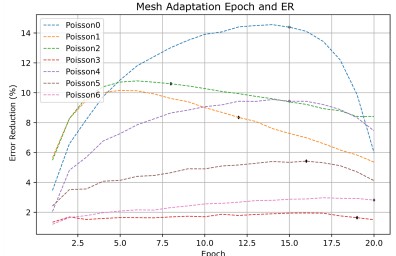

Figure 9: Error reduction in solving the Poisson equation under different adaptive epoch settings.

**Scaling parameter** $\lambda$. To assess the sensitivity of UGM2N to the scaling parameter $\lambda$ in the M-Uniform loss, we conducted an ablation study with $\lambda = 10^{\text{scale}}$ (scale $\in \{-1, 0, 1, 2, 3\}$, 5 runs per value). The left panel of Fig. 10 shows that the converged test loss scales linearly with $\lambda$, consistent with Eq. 8, confirming that $\lambda$ modulates loss magnitude without affecting convergence dynamics. Adaptation performance, measured by error reduction, remains stable across most PDEs (right panel of Fig. 10). Minor sensitivity in Helmholtz1, Poisson4, and Poisson7 (highlighted in red) arises from case-specific optimization challenges, not model limitations, as these cases are difficult across all baselines. Overall, $\lambda$ exerts negligible influence on convergence and adaptation quality, validating the robustness of the M-Uniform loss design.

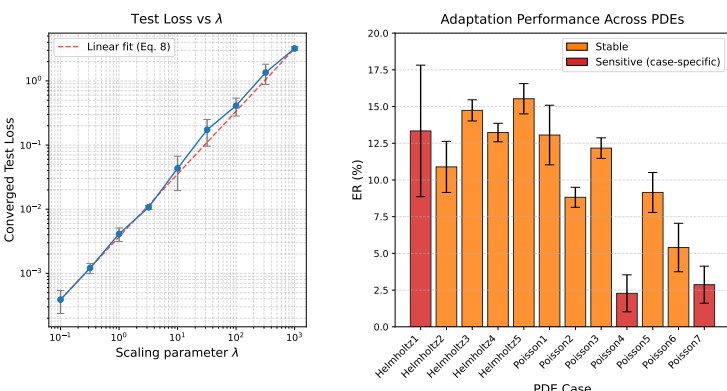

Figure 10: Ablation study on scaling parameter $\lambda$. Left: Converged test loss vs $\lambda$ (log-log scale) with theoretical linear fit (dashed). Right: Error reduction across PDEs; error bars represent ER variability across different $\lambda$ values; red bars highlight sensitive cases.

## 5   Conclusion

We introduce UGM2N, an unsupervised and generalizable mesh movement network. This network removes the requirement for pre-adapted meshes while demonstrating strong generalization capabilities across various PDEs, geometric configurations, and mesh resolutions. By leveraging localized node-patch representations and a novel M-Uniform loss, our approach enforces mesh equidistribution properties comparable to Monge-Ampère-based methods—but in a more efficient, unsupervised manner. Extensive experiments demonstrate consistent performance improvements over both supervised learning baselines and traditional mesh adaptation techniques, achieving significant error reductions without mesh tangling across diverse PDEs and mesh geometries. See App. F for the discussions on limitations and broader impacts.

## Acknowledgment

We appreciate the reviewers for their valuable insights and helpful comments. This research was partially supported by the National Key Research and Development Program of China (2021YFB0300101, 2023YFB3001903), the National Natural Science Foundation of China (12402349), the Natural Science Foundation of Hunan Province (2024JJ6468), and the Youth Foundation of the National University of Defense Technology (ZK2023-11).

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

# A Monge-Ampère-based mesh adaptation

Monge-Ampère (MA)-based mesh adaptation methods leverage optimal transport theory to map gradient or Hessian information from the evolving physical solution onto a mesh density function, thereby generating highly adaptive structured or unstructured meshes that dynamically refine in critical regions. The core idea is to reformulate mesh adaptation as solving the elliptic Monge-Ampère equation:

$$\det(D^2\phi(\mathbf{x})) = \frac{m(\mathbf{x})}{m_0}, \quad \mathbf{x} \in \Omega \tag{10}$$

where $m(\mathbf{x})$ is the monitor function, $m_0$ is a normalization constant, $\phi(\mathbf{x})$ is the convex scalar potential whose gradient defines the optimal coordinate transformation, and $D^2\phi(\mathbf{x})$ is its Hessian matrix. This yields an optimal transport mapping that automatically refines the mesh in high-gradient or high-curvature zones while coarsening it in smooth regions.

However, solving the Monge-Ampère equation demands costly nonlinear iterations—such as Newton or fixed-point methods—that are computationally expensive, sensitive to initial guesses, and often fail to converge in complex domains (e.g., non-convex geometries or those with non-smooth boundaries). These limitations severely restrict the use of MA-based methods in real-time or large-scale adaptive simulations. The present work overcomes this bottleneck by introducing an unsupervised learning framework that accelerates the MA solution process while preserving high-quality adaptive meshes.

# B Method

## B.1 The proposed mesh movement method

The proposed mesh movement method based on UGM2N is presented in Alg. 1. Given an initial mesh and the corresponding flow field, UGM2N iteratively refines the mesh through multiple adaptive operations to obtain the optimal mesh. In this algorithm, operations such as Patch Processing, Mesh Reconstruction, and Convergence Check can all be vectorized, ensuring high computational efficiency in our method.

---

**Algorithm 1** The proposed mesh movement method

---

**Input:** Initial mesh $\mathcal{M}^0 = \{\mathcal{V}^0, \mathbf{U}^0, \mathcal{E}\}$, max epochs $E$
**Output:** Adapted mesh $\mathcal{M}^*$
1: **for** $e = 1$ **to** $E$ **do**
2:     **Patch Processing:**
3:     Construct node patches $\{\mathcal{P}_i\}_{i=1}^N$ from $\mathcal{M}^{e-1}$
4:     **for** each patch $\mathcal{P}_i = \{\mathbf{X}_i, \mathcal{E}_i\}$ **do**
5:         Normalize coordinates $\mathbf{X}_i \to [0,1]^d$    `// Vectorized operations in the loop`
6:         Compute density function $m(\mathbf{x}_i)$ via Eq. 2 and 3
7:         Encode features: $\mathbf{H}_i = \text{NodeEncoder}([\mathbf{X}_i, m(\mathbf{X}_i)])$    `// The operator` $m$ `is applied row-wise to the` $\mathbf{X}_i$
8:         Update positions: $\mathbf{X}_i' = \text{NodeDecoder}(\text{DeformBlocks}(\mathbf{H}_i, \text{EdgeEncoder}(\mathcal{E}_i)))$
9:         Denormalize centering node in $\mathbf{X}_i'$ to the original mesh space
10:     **end for**
11:     **Mesh Reconstruction:**
12:     Assemble adapted mesh $\mathcal{M}' = \{\mathcal{V}', \mathbf{U}', \mathcal{E}'\}$
13:     Interpolate Hessian norm (or grad norm) $\|\mathbf{H}(\mathbf{x}_i')\|$ via Delaunay triangulation
14:     **Convergence Check:**
15:     Compute global uniformity $\mathcal{L}_{\text{var}}(\mathcal{M}')$ via Eq. 9
16:     **if** $\mathcal{L}_{\text{var}}$ stops decreasing or $e == E$ **then**
17:         $\mathcal{M}^* \leftarrow \mathcal{M}'$
18:         **break**
19:     **else**
20:         $\mathcal{M}^e \leftarrow \mathcal{M}'$
21:     **end if**
22: **end for**

---

## B.2 Analysis of M-Uniform loss

Here, we theoretically demonstrate that optimizing the local M-Uniform loss effectively optimizes the objective function associated with mesh equidistribution in one adaptation epoch. Consider a mesh $\mathcal{M}$ (we omit epoch $e$ for clarity), let $L_K \in \{L_{K_j^i} \mid i \in \{1, 2, \ldots, N\}, j \in \{1, 2, \ldots, N_i\}\}$ and $L'_K \in \{L_{K_l} \mid l \in \{1, 2, \ldots, N_e\}\}$ represent discrete random variables defined over mesh elements (see Eq. 6 and Eq. 9), and $I \in \{1, 2, \ldots, N\}$ be a discrete uniform random variable. Here, $N$ denotes the total number of mesh nodes, while $N_i$ indicates the number of mesh elements in the patch centered at node $i$. Simply put, $L'_K$ is a random variable defined on all mesh elements, and $L_{K_j^i}$ is a random variable defined on the mesh elements contained within all patches. According to the law of total variance, we have:

$$\mathrm{Var}(L_K) = \mathbb{E}[\mathrm{Var}(L_K \mid I)] + \mathrm{Var}(\mathbb{E}[L_K \mid I]). \tag{11}$$

The expectation $\mathbb{E}[\mathrm{Var}(L_K \mid I)]$ quantifies the average local variance across node-centered patches, which simplifies to:

$$\mathbb{E}[\mathrm{Var}(L_K \mid I)] = \frac{1}{N} \sum_{i=1}^{N} \left( \underbrace{\frac{1}{N_i} \sum_{l=1}^{N_i} (L_{K_l^i} - \overline{L_{K_l^i}})^2}_{\mathcal{L}_{\mathrm{var}}(P_i)} \right) = \frac{1}{N} \sum_{i=1}^{N} \mathcal{L}_{\mathrm{var}}(P_i). \tag{12}$$

Moreover, from Eq. 9, we have $\mathcal{L}_{\mathrm{var}}(\mathcal{M}) = \mathrm{Var}(L'_K)$. When the samples in $L_K$ are repeated samples from $L'_K$ (i.e., each sample is duplicated $n$ times), we have $\mathrm{Var}(L_K) = \mathrm{Var}(L'_K)$. Assuming that each mesh element appears in approximately the same number of local patches—i.e., the sampling of $L'_K$ in $L_K$ is nearly identical—we can adopt the approximation $\mathrm{Var}(L_K) \approx \mathrm{Var}(L'_K)$. This regularity assumption holds, for example, in high-quality triangular meshes generated by mesh generation software, where almost all mesh nodes have a degree of 6. Moreover, the number of nodes on the boundary is relatively small compared to the number of interior nodes. With $\mathrm{Var}(\mathbb{E}[L_K \mid I]) \geq 0$, we get such an inequality:

$$\mathcal{L}_{\mathrm{var}}(\mathcal{M}) = \mathrm{Var}(L'_K) \approx \mathrm{Var}(L_K) \geq \mathbb{E}[\mathrm{Var}(L_K \mid I)] = \frac{1}{N} \sum_{i=1}^{N} \mathcal{L}_{\mathrm{var}}(P_i) = \frac{1}{\lambda} \mathcal{L}_M(\theta), \tag{13}$$

$$\lambda \mathcal{L}_{\mathrm{var}}(\mathcal{M}) \geq \mathcal{L}_M(\theta). \tag{14}$$

Therefore, $\mathcal{L}_M(\theta)$ provides a valid lower bound for $\lambda \mathcal{L}_{\mathrm{var}}(\mathcal{M})$. In each adaptation epoch, when the model can successfully minimizes $\mathcal{L}_M(\theta)$, it concurrently optimizes the lower bound of $\lambda \mathcal{L}_{\mathrm{var}}(\mathcal{M})$, thereby promoting mesh equidistribution. Moreover, in the limit where $\mathcal{L}_M(\theta) = 0$, all local variances $\mathcal{L}_{\mathrm{var}}(P_i)$ vanish, implying that $L_K$ is constant over all patches. Consequently, $L'_K$ must also be constant, leading to $\lambda \mathcal{L}_{\mathrm{var}}(\mathcal{M}) = 0$, which corresponds to exact mesh equidistribution.

**Convergence and approximation analysis.** The M-Uniform loss optimizes local mesh equidistribution by minimizing $\mathcal{L}_{\mathrm{var}}(\mathcal{P}_i)$ (Eq. 7), similar to traditional mesh smoothing techniques that optimize geometric quality metrics (e.g., aspect ratio). In a serial implementation, sequential node updates improve $\mathcal{L}_{\mathrm{var}}(\mathcal{P}_i)$ for a specific mesh node $i$ at each iteration, suggesting convergence. However, training errors limit exact minimization ($\mathcal{L}_{\mathrm{var}}(\mathcal{P}_i) = 0$), and serial updates are computationally inefficient. Our Jacobi-style parallel updates enhance efficiency but complicate analysis due to data races, update conflicts, and the nonlinear nature of the transformation represented by our model. The non-convex Hessian of the objective function further precludes standard optimization guarantees. Despite these theoretical challenges, we demonstrate through extensive experiments in Section 4 that UGM2N achieves robust convergence.

To further validate convergence, we track the nodal displacement norm $\delta_k = \|\mathbf{X}^{k+1} - \mathbf{X}^k\|$ over iterations, where $\mathbf{X}^k$ denotes the mesh node positions at iteration $k$. As shown in Fig. 11, $\delta_k$ exhibits consistent decay toward zero across five Helmholtz equation cases, confirming convergence. Although $\delta_k$ may stagnate or even increase for larger iterations ($k > 10$) in some instances, our dynamic termination strategy (App. B.1) effectively mitigates this, ensuring stable mesh adaptation.

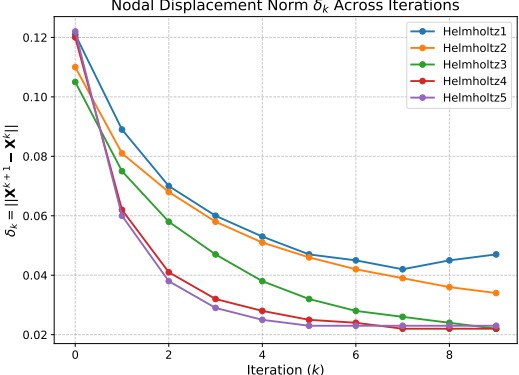

Figure 11: Nodal displacement norm $\delta_k = \|\mathbf{X}^{k+1} - \mathbf{X}^k\|$ across iterations for Helmholtz equations.

## C   Dataset

### C.1   Train data setups

We only used four flow fields on the same mesh to train the model, as shown in Fig. 12. The analytical solutions for the four flow fields are:

1. $u_1(x, y) = 10\sin(2\pi x)\sin(2\pi y)$;
2. $u_2(x, y) = -\frac{5}{\pi}\sin(\pi x)\sin(\pi y)$;
3. $u_3(x, y) = 10(\sin(5x)^{10} + \cos(10 + 25xy)\cos(5x))$;
4. $u_4(x, y) = 10(1 - e^x\cos(4\pi y))$

Additionally, we perform data augmentation on the mesh by introducing random perturbations to the mesh nodes.

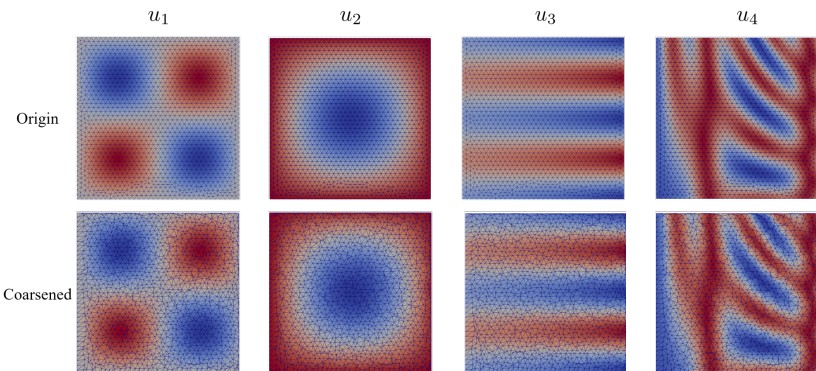

Figure 12: Flow fields for trianing the mesh movement model.

### C.2   Impact of training data volume

UGM2N attains robust generalization using merely 4 flow fields (yielding 10,440 node-patch samples) via its unsupervised learning paradigm. To quantify the effect of training data volume, we evaluated model performance across 4, 8, and 16 flow fields—corresponding to 10,440, 20,880, and 41,760 patches, respectively. As shown in Table 3, performance improves steadily with increasing data volume. Notably, the Poisson and Helmholtz cases exhibit substantial error reductions, improving from 5.21% to 7.43% and 9.94% to 14.21%, respectively. Overall, scaling the training data volume reliably boosts UGM2N's generalization capability, with especially marked gains in complex PDEs, highlighting the critical role of larger datasets in future developments.

Table 3: Error reduction ( %) vs. training data volume (node patches)

| Data Volume | Poisson | Helmholtz | Burgers |
|---|---|---|---|
| 10,440 | 5.21 | 9.94 | 30.07 |
| 20,880 | 6.56 | 13.68 | 29.36 |
| 41,760 | **7.43** | **14.21** | **30.56** |

## C.3   Test data setups

**Helmholtz.** The Helmholtz equation describes the propagation of time-harmonic waves in physics and engineering. Here, we solve an equation of the following form:

$$-\nabla^2 u + u = f, u = g \text{ on } \partial\Omega, \tag{15}$$

where $u$ is the solution variable, $\partial\Omega$ denotes the boundary, $g$ is the boundary function for $u$, and $f$ is the source term. To test the generalization capability of the model, we construct five different solutions (see Table 1), compute $f$ and $g$ to formulate the Helmholtz equation, and evaluate the model's performance.

**Poisson.** The Poisson equation describes how a scalar field responds to a given source distribution, written as $\nabla^2 u = f$. Using the same approach as for the Helmholtz equation, we constructed Poisson equations with seven different exact solutions (see Table 1) to test the model.

**Burgers.** The Burgers equation is a fundamental nonlinear partial differential equation in fluid dynamics, combining convection and diffusion effects. In this paper, we solve the following Burgers equation:

$$\frac{\partial \mathbf{u}}{\partial t} + (\mathbf{u} \cdot \nabla)\mathbf{u} - \nu\nabla^2\mathbf{u} = 0,$$
$$(\mathbf{n} \cdot \nabla)\mathbf{u} = 0 \text{ on } \Omega, \tag{16}$$

where the viscosity coefficient $\nu = 0.005$, and the initial conditions are given in Table 1. The simulation employs a time step of $\Delta t = \frac{1}{30}s$ and runs for a total duration of $0.5s$.

**Airfoil and cylinder flow case.** The airfoil case was simulated under conditions of Mach 0.8 at 1.55° angle of attack, while the cylinder flow case employed a Reynolds number of 100 with characteristic length of 0.2, kinematic viscosity of 0.01, time step of $0.001s$, and total simulation duration of $3.5s$. In our experiments, the airfoil case employed a coarse mesh with 10,466 elements and a high-resolution mesh with 41,364 elements, while the cylinder flow used 5,536 (coarse) and 11,624 (high-resolution) elements.

**Wave equation.** The wave equation is a partial differential equation that describes the propagation of waves (such as sound waves, light waves, water waves, etc.) through a medium or space. This experiment solves the two-dimensional wave equation for the initial value problem on a unit circle, namely:

$$\frac{\partial^2 u}{\partial t^2} - \nabla^2 u = 0, \tag{17}$$

where the initial condition is $u(x, y, 0) = (1 - x^2 - y^2)\sin(\pi x)\sin(\pi y)$. The time step is $0.01s$, and the total time is $2s$. The coarse mesh consists of 2,048 elements, while the high-resolution mesh contains 8,192 elements.

**Moving cylinder and supersonic flow over the wedge.** Both cases are benchmark cases from the Kratos official website [41]. The mesh element count for the moving cylinder case is 9,852, and the element count for the supersonic flow case is 27,115. Please refer to the official site for more details.

## C.4   Monitor function setups

Table 4 presents the values of the monitoring function $\alpha$ for different cases. Notably, in the airfoil case study, the monitoring function employs the gradient norm (rather than the Hessian norm) of mesh nodes to better capture the shock location. To address the long-tail distributions observed in both the cylinder flow and airfoil cases (since most mesh nodes have small Hessian norm values), a logarithmic transformation was applied to the monitoring function at mesh nodes.

Table 4: $\alpha$ for different cases

| Case | $\alpha$ |
|---|---|
| Train data | 5 |
| Poisson, Helmholtz, Burgers, Wave | 5 |
| Cylinder flow, Airfoil case | 10 |

## D  More model training details

We partitioned the dataset into training, validation, and test sets with an 8:1:1 ratio, employing the validation set for early stopping. The model architecture consists of node and edge encoders implemented as two linear layers [2, 512], followed by deformation blocks containing an 8-layer Graph Transformer with 512 hidden dimensions, residual connections, and 4 attention heads, and finally a node decoder comprising a LayerNorm-equipped MLP [512, 256, 2]. All components utilize ReLU activation functions. The model was trained on a machine with an I7-9700KF CPU, NVIDIA RTX TITAN GPU, and 64GB of memory, with the complete model only requiring approximately 3.1 hours of training time.

## E  Results

### E.1  Evaluation metrics

**Error reduction (ER).**  Error reduction quantifies the relative enhancement in PDE solution accuracy, computed as the improvement over the initial coarse mesh solution, where the high-resolution result is treated as the ground truth. For the steady case, error reduction is defined as:

$$\mathrm{ER} = \frac{\|\mathbf{u}_{\mathrm{coarse}} - \mathbf{u}_{\mathrm{ref}}\|_2 - \|\mathbf{u}_{\mathrm{adapted}} - \mathbf{u}_{\mathrm{ref}}\|_2}{\|\mathbf{u}_{\mathrm{coarse}} - \mathbf{u}_{\mathrm{ref}}\|_2} \times 100\%, \tag{18}$$

where $\mathbf{u}_{\mathrm{coarse}}$ is the initial coarse mesh solution, $\mathbf{u}_{\mathrm{ref}}$ is the reference solution on the high-resolution mesh, and $\mathbf{u}_{\mathrm{adapted}}$ is the solution on the adapted mesh. A negative ER value indicates that the mesh adaptation process failed to improve the solution accuracy compared to the initial coarse mesh. For the unsteady case, error reduction is used to evaluate cumulative error reduction, defined as:

$$\mathrm{ER} = \frac{\sqrt{\sum_i^T \|\mathbf{u}_{\mathrm{coarse},i} - \mathbf{u}_{\mathrm{ref},i}\|_2^2} - \sqrt{\sum_i^T \|\mathbf{u}_{\mathrm{adapted},i} - \mathbf{u}_{\mathrm{ref},i}\|_2^2}}{\sqrt{\sum_i^T \|\mathbf{u}_{\mathrm{coarse},i} - \mathbf{u}_{\mathrm{ref},i}\|_2^2}} \times 100\%, \tag{19}$$

where $\mathbf{u}_{\mathrm{coarse},i}$, $\mathbf{u}_{\mathrm{adapted},i}$, and $\mathbf{u}_{\mathrm{ref},i}$ represent the numerical solutions at timestep $i$ from the initial coarse mesh, adapted mesh, and high-resolution mesh respectively.

**Cumulative error.**  Cumulative error refers to the accumulation of errors over time, which is defined as:

$$\text{Cumulative error} = \sqrt{\sum_i^T \|\mathbf{u}_i - \mathbf{u}_{\mathrm{ref},i}\|_2^2}, \tag{20}$$

where $\mathbf{u}_i$ and $\mathbf{u}_{\mathrm{ref},i}$ are the numerical solutions at timestep $i$ from the current mesh and the high-resolution mesh, and $T$ is the total timestep.

**MAE of $C_p$ and $C_D$.**  The MAE (Mean Absolute Error) of the pressure coefficient $C_p$ and drag coefficient $C_D$ measures the errors between the solutions obtained at different positions or time steps and the results from the high-resolution mesh, defined as:

$$C_{p,\mathrm{MAE}} = \mathrm{Mean}_i |C_{p,i} - C_{p,i}^{\mathrm{ref}}|, \quad C_{D,\mathrm{MAE}} = \mathrm{Mean}_i |C_{D,i} - C_{D,i}^{\mathrm{ref}}|, \tag{21}$$

where $C_{p,i}$ and $C_{D,i}$ represent the pressure coefficient and drag coefficient computed on the coarse (adapted) mesh at the $i$-th location (or time step), and $C_{p,i}^{\mathrm{ref}}$ and $C_{D,i}^{\mathrm{ref}}$ denote the corresponding values computed on the high-resolution mesh at the same location/time step.

### E.2  Mesh tangling evaluation on non-convex meshes

To further test whether our model avoids mesh tangling during the adaptation process on non-convex meshes, we adopted the same testing method as UM2N [39]. Specifically, we generated 100 meshes, each containing 8 nodes, with four located at the corners of a unit square and the remaining four randomly sampled within the unit square. The flow field was constructed using a mixture of Gaussian distributions. The results are shown in Table 5, demonstrating that both our model and UM2N consistently produced valid meshes. In contrast, the MA method struggles with non-convex cases and only generates valid meshes in limited scenarios. Fig. 13 shows some meshes after adaptation by our UGM2N model; our model generalizes well to unseen non-convex meshes without producing any mesh tangling.

Table 5: Results of the mesh tangling test on non-convex meshes

| Metric | MA | UM2N | UGM2N |
|---|---|---|---|
| Tangling ratio per mesh (mean±std) | 6.63%±10.91% | **0%** | **0%** |
| Valid mesh | 41 | **100** | **100** |

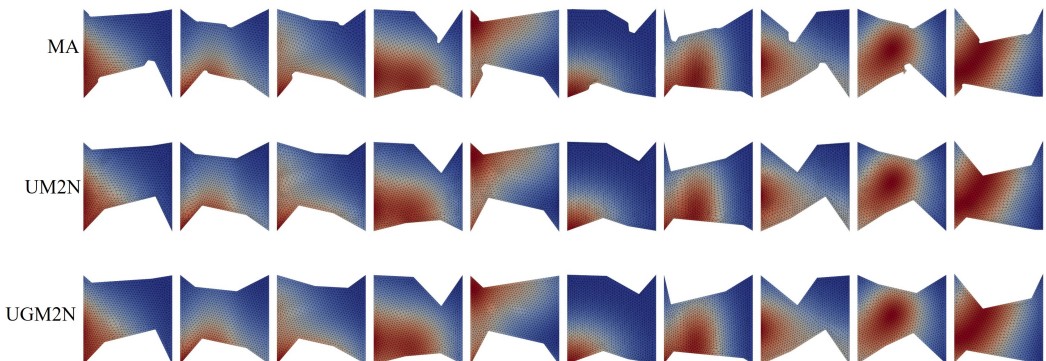

Figure 13: Mesh adaptation results of non-convex meshes (10 out of 100 cases).

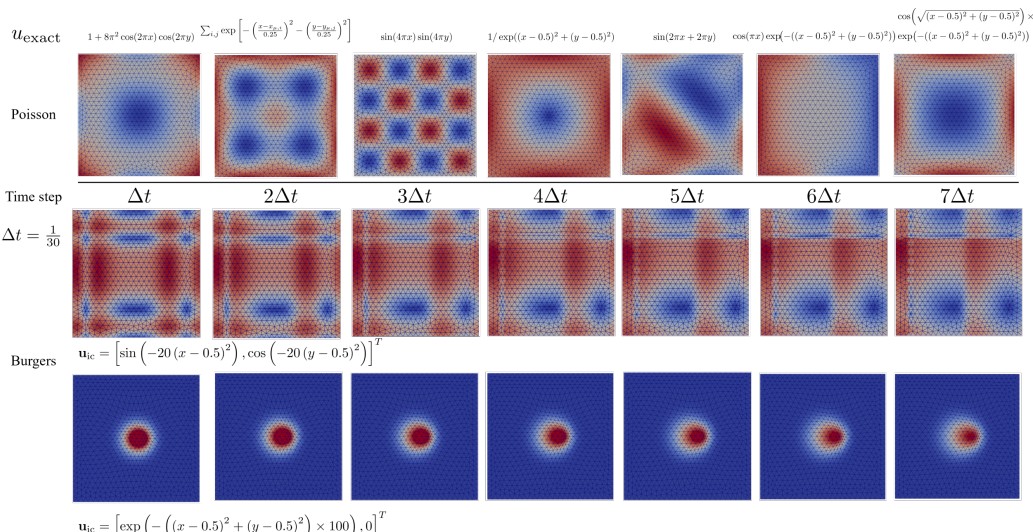

Figure 14: Mesh adaptation on Poisson's equation and Burgers' equation.

## E.3 Mesh adaptation results on Poisson's equation and Burgers' equation

As shown in Fig. 14, our method can effectively generate adaptive meshes in flow fields with different distributions, whether for steady or unsteady cases.

## E.4 Impact of boundary node treatment on model performance

Ensuring geometric consistency of mesh boundaries remains a key challenge in learned mesh movement. The core difficulty arises from inevitable prediction errors in ML models: even minor deviations in predicted boundary node positions can severely distort boundary layers—the thin near-wall regions where viscous, thermal, or electromagnetic gradients are physically dominant and numerically critical.

To further evaluate different boundary handling strategies, we conducted experiments enabling simple clip-to-boundary projection (i.e., projecting adapted boundary nodes back onto the domain boundary if they drift outside), with results summarized in Table 6. The results show that clip projection yields modest gains in internally dominated flows—where gradients or Hessian values are predominantly large in the interior—but delivers substantial improvements when large Hessian values are concentrated near the boundaries while interior values remain small (e.g., Poisson4, Poisson6, Poisson7).

These findings confirm that, while our fixed-boundary strategy ensures stability and prevents boundary layer collapse, simple post-hoc projection can safely enhance performance in scenarios where flow features vary sharply near boundaries. This highlights considerable potential for future improvements through learned boundary constraints or hybrid adaptation-projection schemes, especially in wall-bounded and multi-physics applications.

Table 6: Error reduction with and without boundary clip projection. Cases follow Table 1 in the main paper.

|  | Burgers0 | Burgers1 | Helmholtz1 | Helmholtz2 | Helmholtz3 | Helmholtz4 | Helmholtz5 |
|---|---|---|---|---|---|---|---|
| ER w.o. Clip (%) | 32.22 | 30.19 | 14.11 | 13.15 | 15.03 | 14.09 | 16.98 |
| ER with Clip (%) | 35.27 | 30.18 | 16.68 | 16.17 | 15.94 | 15.57 | 17.34 |
| Gain (%) | ↑3.05 | ↓0.01 | ↑2.57 | ↑3.02 | ↑0.91 | ↑1.48 | ↑0.36 |

|  | Poisson1 | Poisson2 | Poisson3 | Poisson4 | Poisson5 | Poisson6 | Poisson7 |
|---|---|---|---|---|---|---|---|
| ER w.o. Clip (%) | 14.56 | 9.00 | 12.46 | 1.70 | 9.07 | 4.90 | 2.56 |
| ER with Clip (%) | 17.49 | 14.82 | 13.08 | 15.98 | 11.18 | 17.85 | 15.69 |
| Gain (%) | ↑2.93 | ↑5.82 | ↑0.62 | ↑14.28 | ↑2.11 | ↑12.95 | ↑13.13 |

## E.5 The mesh equidistribution condition on the adapted mesh

Fig. 15 shows the absolute error between $\mathcal{L}_K$ and $\overline{\mathcal{L}_K}$ on each mesh element after adaptation, while Table 7 presents the value of $\mathcal{L}_{\mathrm{var}}(\mathcal{M}')$ on the mesh. Quantitative analysis reveals our method produces minimal error values, indicating its adapted mesh most closely approximates the ideal equidistribution condition.

Table 7: $\mathcal{L}_{\mathrm{var}}(\mathcal{M}')$ on the adapted meshes for the Helmholtz equation with different solutions

| Method | Solution of the Helmholtz equation | | | | |
|---|---|---|---|---|---|
|  | $\cos(2\pi y)$ | $\cos(2\pi x)$ | $\cos(2\pi y)\cos(2\pi x)$ | $\cos(2\pi y)\cos(4\pi x)$ | $\cos(4\pi y)\cos(2\pi x)$ |
| MA [38] | 3.21e-7 | 3.89e-7 | 1.68e-7 | 1.30e-7 | 1.50e-7 |
| M2N [9] | 4.41e-7 | 5.90e-7 | 2.62e-7 | 4.67e-7 | 4.82e-7 |
| UM2N [40] | 5.41e-7 | 5.50e-7 | 2.89e-7 | 3.32e-7 | 3.73e-7 |
| UGM2N (ours) | **2.12e-7** | **2.68e-7** | **1.52e-7** | **1.10e-7** | **1.19e-7** |

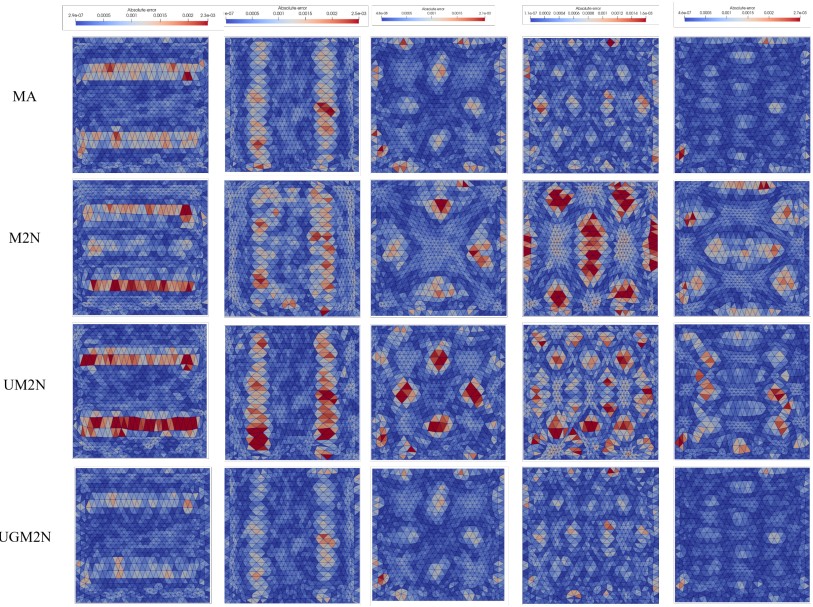

Figure 15: $|\mathcal{L}_K - \overline{\mathcal{L}_K}|$ on the adapted meshes.

## E.6    Detailed efficiency analysis and scalability

A key consideration is the scalability of our model to large meshes. Theoretically, our GNN-based model imposes no restrictions on mesh scale, as it can process graphs of arbitrary size. However, practical GPU memory constraints limit single-GPU performance. On an RTX TITAN GPU (24GB), UGM2N can process up to 40,000 mesh elements, with inference requiring 345ms and interpolation 61ms, achieving a significant speedup over the MA method (25,510ms).

To further evaluate scalability, we measured inference, interpolation, and I/O times (data transfer between CPU and GPU) across varying mesh sizes (Table 8).It can be observed that I/O operations account for the majority of the time overhead. This is because, in each epoch, the mesh must be transferred from the GPU back to the CPU for Delaunay triangulation-based interpolation. If a GPU-accelerated Delaunay triangulation algorithm were available, both inference and interpolation could be performed entirely on the GPU—this represents one of the key directions for future optimization.

Furthermore, UGM2N's absence of data dependencies between node patches enables efficient parallelization across $N$ GPUs, with processing time scaling as $O(1/N)$ (excluding communication overheads). This supports real-time mesh adaptation for large-scale simulations.

Table 8: Timing breakdown (ms) for UGM2N across mesh sizes. For each element count, we repeated the experiment ten times and report the mean and standard deviation.

| Elements | Inference | Interpolation | I/O |
|---|---|---|---|
| 1,478 | 14.3±0.12 | 0.75±0.02 | 48.7±0.84 |
| 2,396 | 17.9±0.12 | 1.30±0.30 | 75.9±0.85 |
| 4,126 | 25.5±0.45 | 2.49±0.02 | 136±0.39 |
| 9,248 | 43.9±0.67 | 7.17±0.07 | 302±0.83 |
| 36,098 | 354±1.88 | 60.0±0.60 | 1,010±1.55 |

## E.7    The mesh adaptation results on the irregular and anisotropic mesh

In App. B.2, we derive the validity of the proposed loss function by assuming mesh regularity $\mathrm{Var}(L_K) \approx \mathrm{Var}(L'_K)$. This assumption holds for meshes generated by mature mesh generation software, as they often ensure a certain level of mesh quality (e.g., for a 2D test case with 60k mesh

elements generated by mature mesh generation software, we observed that 92.96% of nodes had a degree of 6, while 2.96% had a degree of 5—meaning around 95% of nodes fell into these two categories). Furthermore, we tested the model's applicability to irregular and anisotropic meshes—i.e., scenarios where the assumption does not hold.

**Irregular mesh.** We uniformly sampled mesh points within a unit rectangular domain and generated meshes using Delaunay triangulation, followed by mesh smoothing to optimize quality. As shown in Fig. 16, the resulting meshes feature numerous nodes with varying degrees, thus failing to satisfy the regularity assumption. We generated a total of 10 such samples, with each sample's flow field solution defined as $\cos(2\pi(x - \mu_x))$, where $\mu_x$ is a random value uniformly drawn from $[0, 1]$. The error reduction results for our model are presented in Table 9. These results demonstrate that, despite being trained on regular meshes, our model generalizes effectively to highly irregular meshes, achieving substantial error reduction in 8 out of 10 cases.

Table 9: Error reduction of UGM2N on irregular meshes

|         | Case 1 | Case 2 | Case 3 | Case 4 | Case 5 | Case 6 | Case 7 | Case 8 | Case 9 | Case 10 |
|---------|--------|--------|--------|--------|--------|--------|--------|--------|--------|---------|
| **ER (%)** | -2.07 | 16.07 | 10.69 | 11.11 | 18.10 | 14.24 | -2.10 | 8.46 | 9.84 | 7.16 |

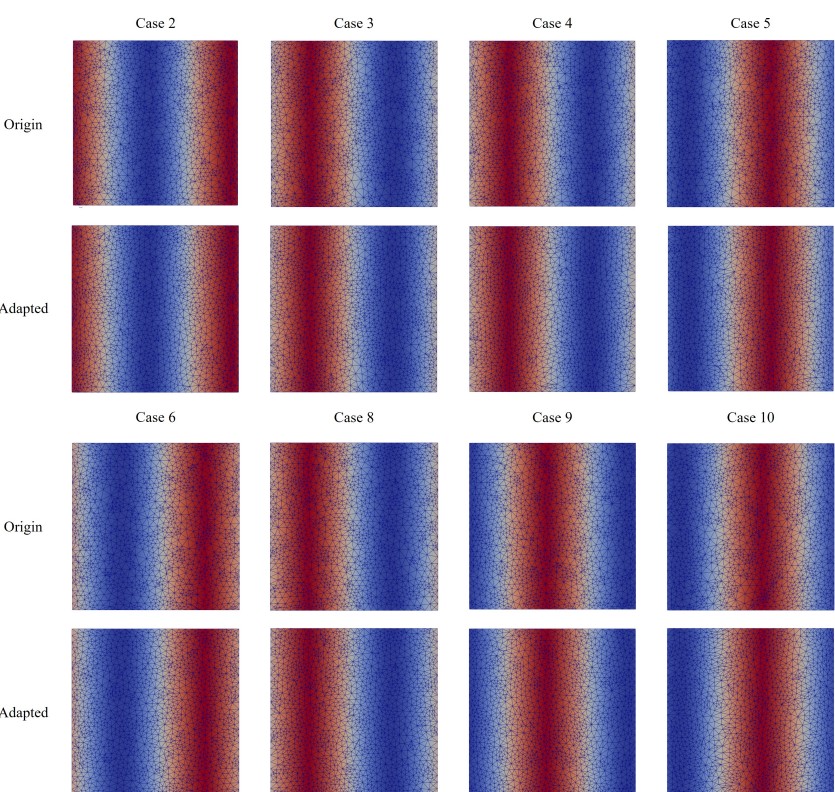

Figure 16: Mesh adaptation results on irregular meshes (successful cases). It can be observed that mesh nodes move toward regions of high Hessian values in the flow field.

**Anisotropic mesh.** To evaluate our model's performance on anisotropic meshes, we solve the Poisson equation with the exact solution $u(x, y) = (1 - e^{\frac{-(x-0.5)^2}{0.01}})((x - 0.5)^2 - 1)$ for $[x, y] \in [0, 1] \times [0, 1]$. We first generate preliminary stretched mesh elements via anisotropic adaptation to produce the initial anisotropic mesh, as shown in Fig. 17. This mesh is then input to the mesh adaptation models, which further optimize the node distribution to align with the solution's characteristics. In this solution, the Hessian values peak at approximately $x = 0.3$, $0.5$, and $0.7$. As illustrated in Fig. 17, compared to the initial anisotropic mesh, both the UM2N and UGM2N models more accurately capture these three locations, while the MA method yields an invalid mesh. The absolute error reductions achieved after

solving with the optimized anisotropic meshes are presented in Table 10. These results demonstrate that our model achieves the optimal error reduction, confirming its applicability to anisotropic meshes.

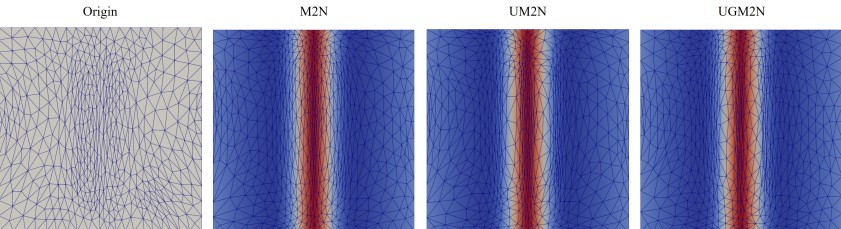

Figure 17: Mesh adaptation results on Anisotropic meshes. The UGM2N model successfully captures the peak locations of the Hessian values.

Table 10: Absolute error reduction on anisotropic mesh

|  | MA | M2N | UM2N | UGM2N |
|---|---|---|---|---|
| **Absolute error reduction** | Fail | Fail | 0.02 | **0.09** |

### E.8 Large-scale robustness validation on random geometries

To rigorously evaluate generalization under extreme geometric and flow variability, we conducted a large-scale experiment on 1,000 randomly generated mesh samples. Each sample features a random polygon with 3–6 edges and a synthetic flow field drawn from a Gaussian mixture model. We solved the Helmholtz equation using these flow fields as exact solutions and compared mesh adaptation performance across methods.

Results are presented in Table 11 and Fig. 18. Both MA and UM2N exhibit high variance and frequent negative error reduction, reflecting poor generalization across diverse geometries and flow structures. In contrast, UGM2N achieves a Positive ER Ratio of 0.807 (807 successful cases out of 1,000) with a stable mean ER of 13.99% ± 21.05%, demonstrating superior robustness. M2N was excluded from comparison due to its inability to generalize to arbitrary geometries and flow fields without case-specific training.

Table 11: Robustness comparison on 1,000 random polygonal domains with Gaussian mixture flow fields (Helmholtz equation). Positive ER Ratio = fraction of cases with ER > 0.

| Method | ER (mean% ± std%) | Positive ER Ratio |
|---|---|---|
| MA | -227.76 ± 468.82 | 0.110 |
| UM2N | -50.72 ± 68.36 | 0.245 |
| UGM2N (Ours) | **13.99 ± 21.05** | **0.807** |

### E.9 More results of qualitative experiments

For the supersonic flow and moving cylinder cases, MA method exhibited divergence during the solution of the monitor function equation in both cases, ultimately failing to produce valid adapted meshes. As shown in Fig. 19, the M2N method displayed inconsistent behavior, particularly in the moving cylinder case, where it generated well-adapted meshes at certain time steps but yielded highly distorted elements at others. For the supersonic flow, M2N successfully captured the shock position; however, the resulting mesh exhibited lower nodal density at the shock compared to UGM2N. Similarly, UM2N identified the shock location in the supersonic case but introduced undesirable element distortions in irrelevant regions away from the shock. In the moving cylinder scenario, while UM2N effectively fitted the flow field distribution, its excessive adjustments led to prominent mesh distortions.

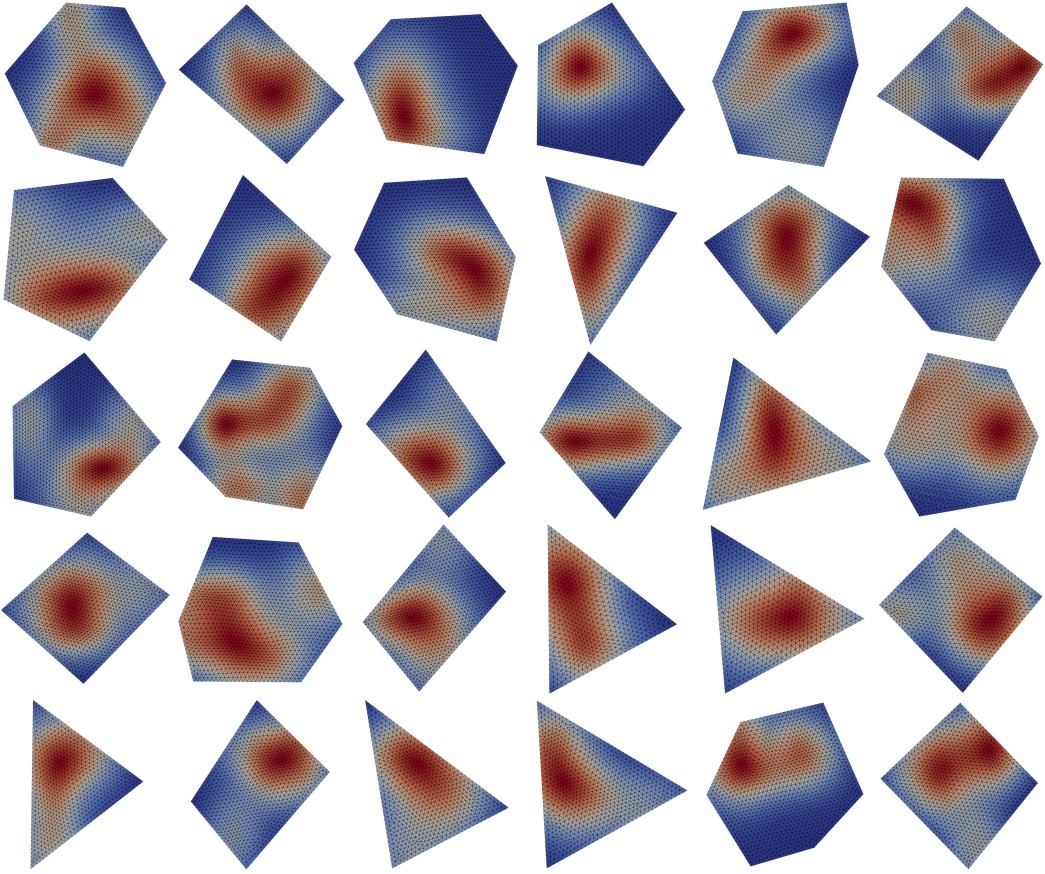

Figure 18: Mesh adaptation results of UGM2N on different flow fields across different meshes (30 cases out of 1000). Whether it is changes in the mesh or changes in the flow field, UGM2N demonstrates excellent robustness.

## F   Limitations and broader impacts

**Limitations.**  1) The model currently fixes boundary nodes during adaptation to ensure geometric consistency and prevent boundary layer distortions; while post-hoc clip projection to the boundary yields modest improvements in some cases, future work could integrate learned boundary constraints for more flexible handling. 2) The approximation in Eq. 13 assumes mesh regularity (i.e., similar node degrees), but empirical tests on irregular meshes (App. E.7) demonstrate robust performance in 80% of cases despite violations. Further validation on highly non-uniform or anisotropic meshes with extreme degree variations remains warranted. 3) The model adopts a relatively simple and lightweight architecture. Future work could explore more complex model designs to achieve better mesh adaptation performance. 4) By adopting a node-patch-based processing strategy, our method achieves scale invariance and translation invariance—that is, adaptation results remain unchanged under uniform scaling or translation of the mesh. However, we observe that all AI-based mesh adaptation methods currently lack rotation invariance: rotating the coordinate system of the mesh leads to inconsistent results. This remains an important open challenge for future work.

**Broader impacts.**  The UGM2N method proposed in this paper significantly improves the efficiency of mesh movement techniques in mesh adaptation through unsupervised learning and localized mesh adaptation technology. It reduces the high computational costs of traditional mesh adaptation methods, thereby accelerating the simulation of complex physical phenomena such as fluid dynamics and heat transfer under limited computational resources. Its generalizability allows it to be applied to various partial differential equations and geometric shapes, enhancing the practical engineering applications of current AI-based mesh adaptation methods.

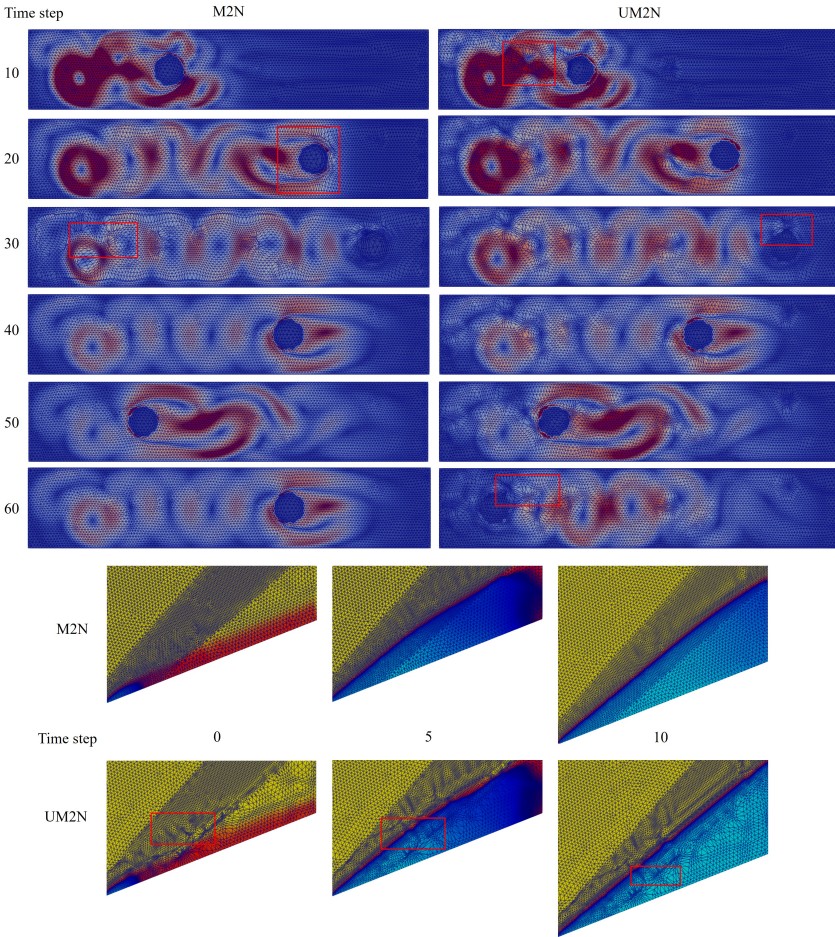

Figure 19: Mesh adaptation results of the M2N and UM2N models for the supersonic flow and moving cylinder cases. Red boxes mark the locations of mesh distortions.

