# OpenReview forum: "UGM2N: An Unsupervised and Generalizable Mesh Movement Network via M-Uniform Loss"
_NeurIPS.cc/2025/Conference — NeurIPS 2025 poster_

### Official Review · Reviewer_YRmJ · 2025-06-29

**Clarity:** 3
**Significance:** 3
**Originality:** 4
**Rating:** 3
**Confidence:** 4

**Summary:**

This paper presented an unsupervised mesh movement network that performs excellent generalizability across various PDEs and designed a M-uniform loss to constrain the equidistribution within mesh nodes. The method, named UGM2N, leverages a localized, patch-based approach inspired by vision transformers, which allows it to adapt meshes without requiring pre-adapted examples for supervision.

**Questions:**

**model's true geometric invariance**. In Table 1, the Helmholtz equation test cases for the solutions cos(2πy) and cos(2πx) yield different Error Reduction (ER) values of 14.11% and 13.15%, respectively. Given that these two solution fields are mathematically symmetric. Could the authors elaborate on the potential cause of this discrepancy? Is this slight performance difference a result of inherent asymmetries in the training dataset, the underlying mesh triangulation, or another artifact of the learning process?

**Ethical Concerns:**

["NO or VERY MINOR ethics concerns only"]

**Limitations:**

1. lack of real-world scenarios benchmarks.
2. the writing of this paper is hard to follow.

**Quality:**

3

**Strengths And Weaknesses:**

Strenth:

1. **Strong experimental results showing excellent generalizability across various PDEs**.
2. **Neat architectral design**. The decision to process local node patches instead of the entire mesh is an elegant and effective design choice.

Weakness:

1. **Clarity of key formulations.** The mathematical exposition could be improved for clarity. For instance, in Section 3.1, Equation (1) introduces a function $\rho(x)$ within the integral definition of $σ =∫_K ρ(x) dx$. However, the function $\rho(x)$ is never defined or explained in the text.
2. **The theoretical justification for the M-Uniform Loss is not fully established.** In Appendix A.2, the proof that the proposed loss $L_M(θ)$ acts as a lower bound for the global mesh variance $L_{var}(M)$ hinges on the assumption that $Var(L_K) ≈ Var(L'_K)$ (line 465). This is justified by stating that each mesh element appears in a similar number of local patches, a condition that holds for regular meshes but is less certain for the highly irregular or complex geometries in real-world scenarios.
3. **Lack of validation in large-scale, real-world scenarios.** The experiments (e.g., NACA airfoil, cylinder flow), are confined to 2D problems with relatively small mesh sizes (up to ~10k elements). Its performance on complex, industrial-scale 3D simulations, which often involve millions of elements remains untested.

---

> ### Author Rebuttal · Authors · 2025-07-31
>
> We sincerely thank the reviewer YRmJ for their valuable comments and suggestions, which help significantly improve the quality of this manuscript. Below, we provide a point-by-point response to your feedback, with annotations indicating whether each item pertains to a **Question (Q)** , **Weakness (W)** , or **Limitation (L)** in our original submission.
>
> ### **Response to W1\L1**
>
> **Writing and Typo Issues.**
>
> We sincerely appreciate your feedback. In the expression $σ =∫_K ρ(x) dx$, the function $\rho(x)$ should actually be $m(x)$. We apologize for this notational error in our manuscript. For the clarity of our writing，we recognize that certain sections may have been overly technical or insufficiently structured, which could hinder readability. To address this concern, we will take the following steps in revision:
>
> 1. **Improved Organization** – We will refine the logical flow of ideas, ensuring a clearer progression from motivation to methodology and results.
> 2. **Enhanced Explanations** – Key concepts and assumptions will be expanded with additional intuitive examples or illustrative diagrams where appropriate.
> 3. **Language Edits** – Complex sentences will be simplified, and terminology will be consistently defined to improve accessibility without sacrificing rigor.
>
> We are committed to making the paper more accessible to a broader audience while maintaining its technical depth. Please let us know if specific sections require further attention, and we would be glad to provide additional clarifications.
>
> ### **Response to W2**
>
> **Justification of the assumption.**
>
> Thank you for raising this insightful question.  In our M-Uniform loss, we introduced $Var(L_K) ≈ Var(L'_K)$ as a mild assumption. Since our goal is to handle triangular meshes, we also assume that our meshes possess a certain level of quality, which is achievable with modern mesh generation software—even for highly irregular or complex geometries.
>
> In fact, our requirement that *each mesh element appears in a similar number of local patches* essentially means that each mesh node should have a similar degree (i.e., the number of connections to neighboring nodes). Generally, boundary nodes are relatively few compared to interior nodes, and most interior nodes have a degree of 6 or 5 (with a small fraction being 4, which already indicates a more distorted element). In a 2D test case with 60K mesh elements generated by mature mesh generation software, we observed that 92.96% of nodes had a degree of 6, while 2.96% had a degree of 5—meaning around 95% of nodes fell into these two categories. Therefore, we consider this a reasonable assumption.
>
> To further demonstrate our model's performance when this assumption does not hold, we have tested it on real-world cases with arbitrary geometry boundaries, specifically in the Tohoku tsunami simulation (ocean modeling), where only about 69% of nodes had degrees of 6 or 5. The results showed that our model could still accurately capture wave propagation, with mesh element sizes dynamically adapting to wave propagation. This also highlights our model's generalization capability for highly irregular or complex geometries. Additionally, we have evaluated our model's performance on highly irregular meshes where mesh elements do not appear in a similar number of local patches. The flow field solution for each sample was defined as $\cos(2  \pi  (x - \mu_x))$, where $\mu_x$ is a random value in [0, 1]. The error reduction results achieved by our model are presented below:
>
> |            | Case 1 | Case 2 | Case 3 | Case 4 | Case 5 | Case 6 | Case 7 | Case 8 | Case 9 | Case 10 |
> | ---------- | ------ | ------ | ------ | ------ | ------ | ------ | ------ | ------ | ------ | ------- |
> | **ER (%)** | -2.07  | 16.07  | 10.69  | 11.11  | 18.10  | 14.24  | -2.10  | 8.46   | 9.84   | 7.16    |
>
> The experimental results demonstrate that even when mesh elements do not appear in a similar number of local patches, our model can still achieve accuracy improvement in most cases(achieving success in 8 out of 10 cases).
>
> ### **Response to W3\L2**
>
> **Real-world scenarios.** Thank you for your suggestions. Based on your feedback and Reviewer AXEC’s comments, we have added a **real-world case with arbitrary geometry boundaries**, sourced from Thetis project in ocean modeling. The experimental results demonstrate that our model dynamically adapts mesh elements to accurately simulate wave propagation while maintaining high-quality, distortion-free meshes without tangling, even under complex boundary conditions.
>
> Due to technical constraints (e.g., one case requiring additional data access and another involving website maintenance), we were unable to include the other two real-world scenarios (**Columbia River plume simulation and North Sea tidal simulation**) at this stage. However, we will make every effort to incorporate all three cases in the revised manuscript to further validate the robustness of our method.
>
> **Validation in 3D large-scale scenarios**
>
> Thank you for your feedback. Theoretically, our method imposes no restrictions on mesh scale, as our GNN-based model can accept graphs of arbitrary size as input. However, in practice, this often leads to exceeding GPU memory limits.
>
> On a single RTX TITAN GPU with 24GB memory, our model can handle approximately 40,000 mesh elements, with model inference taking about 345ms and interpolation requiring around 61ms per epoch. In comparison, the MA method takes approximately 25510ms, demonstrating that our approach offers significant efficiency advantages over the MA method. We have also verified the proportion of model inference recommendation, interpolation time, and I/O time under different numbers of mesh elements, and the results are as follows. It can be seen that I/O time is the main performance bottleneck, which inspired our future optimization direction.
>
> | Element number | Infer. time (ms) | Inter. time (ms) | IO. time (ms) |
> | - | - | - | - |
> | 1478           | 14.3±0.12        | 0.748±0.015      | 48.7±0.84     |
> | 2396           | 17.9±0.12        | 1.30±0.30        | 75.9±0.85     |
> | 4126           | 25.5±0.45        | 2.49±0.021       | 136±0.39      |
> | 9248           | 43.9±0.67        | 7.17±0.070       | 302±0.83      |
> | 36098          | 354±1.88         | 60.0±0.60        | 1010±1.55     |
>
> On the other hand, our method is highly suitable for parallelization because there are *no data dependencies* between individual node patches. This allows for easy distribution across N GPUs to enable large-scale mesh processing. Additionally, we can trade time for space by processing patches of mesh points incrementally, thereby increasing the manageable data scale. Adopting this approach (without considering additional overheads like patch grouping and aggregation), our model's processing time for meshes on the order of 1 million elements is roughly 10s, which still outperforms the MA method (which already takes 25s for 40,000 elements).
>
> ### **Response to Q**
>
> **Geometric rotation invariance property.**
>
> We sincerely appreciate the reviewer for raising this intriguing question. First, we attribute these minor discrepancies to inherent asymmetries in the underlying mesh triangulation—rotating the test mesh by 90° fails to achieve perfect overlap with the original mesh. More importantly, this observation prompts a fundamental investigation into whether intelligent mesh adaptation methods possess rotation invariance: does rotating the mesh (while preserving nodal flow variables) affect error reduction performance?
>
> In principle, arbitrary angular rotations should not influence error reduction when flow field variables remain unchanged at mesh nodes. However, the black-box nature of neural networks makes this difficult to guarantee theoretically. To empirically validate this property, we have conducted systematic experiments rotating both the Helmholtz case mesh and its flow field through 360° (in 10° increments), evaluating error reduction across all angles for different ground truth solutions. The comprehensive results are presented below:
>
> | PDE Type   | MA               | M2N             | UM2N         | UGM2N        |
> | - | - | - | - | - |
> | Helmholtz1 | 15.60 ± 0.0001   | -16.42 ± 16.54  | 12.03 ± 0.76 | 12.25 ± 4.32 |
> | Helmholtz2 | 10.30 ± 0.0079   | -27.72 ± 10.02  | 8.16 ± 1.45  | 10.64 ± 3.79 |
> | Helmholtz3 | 13.48 ± 0.0000 | -17.54 ± 7.75   | 6.58 ± 1.08  | 12.87 ± 3.38 |
> | Helmholtz4 | 10.87 ± 0.0000 | -282.57 ± 49.40 | -0.92 ± 1.56 | 11.92 ± 1.86 |
> | Helmholtz5 | 13.50 ± 0.0000 | -242.00 ± 64.94 | 4.67 ± 1.66  | 14.33 ± 2.34 |
>
> The results clearly demonstrate that, compared to the MA method, intelligent mesh adaptation approaches exhibit poorer rotation invariance, as evidenced by their larger performance variance ranges. This insight suggests a valuable direction for future research to enhance geometric invariance in learning-based adaptation methods.
>
> While our method shows inferior rotation invariance compared to UM2N, we emphasize that our node-patch processing framework inherently possesses two critical advantages:
>
> 1. **Scale invariance**: The adaptation results remain unchanged under uniform scaling of the mesh.
> 2. **Translation invariance**: The output are consistent  under arbitrary translational transformations of the grid.
>
> These properties, which M2N and UM2N fundamentally lack, provide significant practical benefits in real-world applications where mesh scales and positions may vary. The limitation in rotation invariance remains an open challenge that requires further investigation in the future work.
>
> ------
>
> Thank you very much for your constructive and insightful feedback. We have carefully considered each point and provided detailed responses. We welcome any further comments and look forward to the next stage of the review process.

---

> ### Author Response · Authors · 2025-08-06
> **Invitation to Join Discussion – NeurIPS 2025 Submission #12196: "UGM2N: An Unsupervised and Generalizable Mesh Movement Network via M-Uniform Loss"**
>
> Dear Reviewer YRmJ,
>
> I hope this message finds you well.
>
> We would like to sincerely thank you for your initial review of our submission to NeurIPS 2025 (Paper ID: 12196, Title: "UGM2N: An Unsupervised and Generalizable Mesh Movement Network via M-Uniform Loss"). We truly appreciate the time and effort you have already invested in evaluating our work.
>
> Following the rebuttal phase, several of the reviewers engaged in further discussion and updated their assessments considering our responses. We are grateful for the constructive feedback and the opportunity to clarify key points.
>
> We noticed that you have not yet participated in the post-rebuttal discussion. If you are available and willing, we would be deeply grateful if you could join the conversation and share any additional thoughts or questions you may have in light of the authors’ response. Your perspective is highly valued and would be instrumental in helping the AC and the committee reach a well-informed decision.
>
> Thank you again for your time and consideration. We fully respect your schedule and judgment, and we appreciate any further input you might be able to provide.
>
> Warm regards,
>
> The Authors.

---

> ### Comment · Reviewer_YRmJ · 2025-08-07
>
> Thank you for your thorough rebuttal. I would like to keep the original score unchanged.

---

> > ### Author Response · Authors · 2025-08-08
> >
> > Dear Reviewer YRmJ,
> >
> > I hope you are doing well. As the discussion period is drawing to a close, we were wondering if you might be able to share whether our rebuttal has addressed your concerns. If some points have been clarified while others remain unresolved, we would be happy to provide further explanation or make improvements. We understand that your time is valuable, and we sincerely appreciate your consideration.

---

> > > ### Comment · Reviewer_YRmJ · 2025-08-09
> > >
> > > Yes. I think many concerns are explained not resolved like the one in symmetry. If you can't explain that theoretically, how could I buy the whole story? So I would like to keep my score unchanged.

---

> > > > ### Author Response · Authors · 2025-08-09
> > > >
> > > > Thank you for your response. As we stated in the rebuttal regarding symmetry:
> > > >
> > > > "We attribute these minor discrepancies to inherent asymmetries in the underlying mesh triangulation—rotating the test mesh by 90° fails to achieve perfect overlap with the original mesh."
> > > >
> > > > The observed differences arise from the rotational asymmetry of the mesh itself. When the meshes differ, the underlying problems being solved are inherently distinct—even if their solutions appear similar. Theoretically, identical results cannot be expected in such cases, even with professional solvers.
> > > >
> > > > Please let us know if you have any further questions; we’d be happy to clarify.

---

> > > > ### Author Response · Authors · 2025-08-09
> > > >
> > > > Regarding the justification of the assumption, as mentioned in our rebuttal, we believe our assumption is reasonably valid for the meshes involved in practical simulations, which is supported by empirical evidence. We have also tested the model’s performance when the assumption does not hold, and the results remain favorable.
> > > >
> > > > For real-case simulations, we have added additional experiments to evaluate the model’s performance, and the results further demonstrate the effectiveness of our method.
> > > >
> > > > Concerning large-scale mesh processing, to the best of our knowledge, handling million-scale meshes remains a significant challenge for AI-based CFD method. Our work primarily focuses on proposing an unsupervised and generalizable mesh adaptation model rather than high-performance applications. In the rebuttal, we also discussed the scalability of our approach—our design can be easily extended to large-scale meshes.
> > > >
> > > > We hope this addresses your concerns.

---

> ### Author Response · Authors · 2025-08-07
>
> Dear Reviewer YRmJ,
>
> First and foremost, we would like to express our sincere gratitude for your valuable review comments. The issues you raised in your review report are highly insightful and have significantly contributed to improving our work. We have provided detailed responses to your concerns in our rebuttal.
>
> In the rebuttal, we have made additional clarifications regarding the points you highlighted—such as the clarity of our formulations, the justification for the assumptions in the M-Uniform Loss, and experiments under real scenarios. Furthermore, we have included three new experiments to: (1) validate the reasonableness of the assumptions in M-Uniform Loss (Response to W2), (2) address the efficiency of our method (Response to W3\L2), and (3) discuss the geometric invariance of our intelligent mesh adaptation approach (Response to Q). **We would greatly appreciate it if you could let us know whether our responses have adequately addressed your concerns.**
>
> We deeply value your professional judgment, and with the utmost respect, we are writing this follow-up message in the hope of gaining a deeper understanding of any remaining concerns—particularly aspects that may not have been sufficiently clarified in our rebuttal—and to explore whether there might be further opportunities for improvement to help our paper meet the acceptance criteria.
>
> Thank you once again for your time and thoughtful feedback.
>
> Best regards,
>
> The Authors

---

### Official Review · Reviewer_dnRz · 2025-06-29

**Clarity:** 3
**Significance:** 2
**Originality:** 1
**Rating:** 3
**Confidence:** 5

**Summary:**

The authors present an unsupervised approach to adaptive mesh relocation using a residual graph transformer based mesh deformer. Trained with a patching strategy inspired by vision transformers and using a loss that equidistributes a monitor function inspired by Monge-Ampere. An iterative meshing strategy is used to push the mesh towards a global equidistribution equilibrium. Evaluation is performed on a range of PDE problems Poisson, Helmholtz, Burgers.

**Questions:**

N/A

**Ethical Concerns:**

["NO or VERY MINOR ethics concerns only"]

**Final Justification:**

The authors have addressed my concerns on originality and time complexity. I update my too low score but remain borderline reject due to the limited extent of the novelty on the UM2N baseline.

**Limitations:**

Yes

**Quality:**

3

**Strengths And Weaknesses:**

Novelty: I am concerned the main novelty of this work, the equidistributional loss function, coincides with a work that may be classifiied as a contemporaneous under the reviewer guidlines (see point 6. of https://neurips.cc/Conferences/2020/PaperInformation/ReviewerGuidelines). The authors cite Rowbottom et al "G-Adaptivity: optimised graph-based mesh relocation for finite element methods". They note that this work directly optimises against FEM error but miss that their $L_{equi}(Z)$ in equation (9) proposes the same equidistribution regularizer which is the main focus of this work in equation (7). Rowbottom et. al also use a patching based strategy inspired by vision transformers for scaling their approach. It should be noted however there are slight differences, Rowbottom et. al do not train solely on the equidistribution loss and in addition train on the full FEM truncation error. Moreover Rowbottom et. al use a global loss regulazier using a gradient based monitor function versus the current work that uses a local patch based Hessian monitor. A key detail is that the authors cite Rowbottom et. al July 2024 however a more recent version was uploaded to arxiv 6 Feb 2025 and is now public on ICML 2025 proceedings - https://icml.cc/virtual/2025/papers.html?search=G-Adaptivity:+optimised+graph-based+mesh+relocation+for+finite+element+methods. The work in July 2024 did not contain the equidistribution regularizer and based on the reviewer guidance, the arxiv-only work would be classified as "non-refereed work" and the ICML publication announcement falls inside the 2 months window.

I would ask the AC to determine if the authors should be aware of this updated work, if so, then the submission lacks the sufficient novelety to be accepted at NeurIPS however if the determination is that this work need not be referenced then below I include my review of the paper on it's own merrit.

The proposed equidistributional loss is a powerful adaption, it allows GNN meshing approaches to be trained unsupervised without the require of expert meshes and aligns theoretically with classical Monge-Ampere methods. This can be seen by much stronger results compared to the supervised baselines M2N and UM2N and is further evidenced in the loss function ablation in Table 2. The novel patching strategy inspired by vision transformers allows for scaling to large meshes by paralysing over the domain.

The iterative mesh adaption may be a useful contribution allowing the mesh deformation to flow until convergence of equidistribution, resembling the parabolic relaxation of Monge Ampere methods. However there are some problems with it's presentation, it is clear how this is applied in inference, but in training it seems there is a distribution shift such that the mesh initialisation for later epochs approaches the target mesh deformation. The potential risk of training distribution shift is not discussed.

In evaluation Meshing time is not reported, which is common for baselines M2N/UM2N. This may be because the interative procedure involving interpolation is costly.

In the abstract the authors claim "guaranteed error reduction without mesh tangling" but I could not find any further discussion or expansion of this claim.

---

> ### Author Rebuttal · Authors · 2025-07-31
>
> ### **Response to Concerns Regarding Novelty and Originality**
>
> Thank you for your detailed evaluation and thoughtful comments on our manuscript. We appreciate your concerns regarding the novelty of our contributions and would like to clarify the originality of our approach through a point-by-point response.
>
> First, we acknowledge the relevance of the recent work by Rowbottom et al. and appreciate the opportunity to clarify the citation timeline. Our research began in September 2024, during which the latest available version of their paper was the July 2024 preprint, which we cited. At that time, the preprint did not include elements overlapping with our locally-developed M-Uniform loss function.  The ICML 2025 version was announced as accepted in May 2025 (the final published version will be available even later), and we only became aware of its final content after submitting our manuscript. Since then, we have carefully reviewed the latest version of Rowbottom et al.'s work.  Rowbottom et al.'s work represents a significant and timely contribution to the field, demonstrating impressive theoretical depth and empirical rigor. Their integration of GNNs with Firedrake to minimize finite element solution errors and the Graph DiffFormer architecture represent a notable contribution, particularly in addressing mesh entanglement.  We will ensure proper citation  of this work in our revised manuscript.
>
> Second, our two key innovations - the M-Uniform Loss and node-patch based mesh adaptation approach - remain fundamentally distinct from Rowbottom et al.'s work. As you noted, our method focuses entirely on local node optimization, while they continue using M2N/UM2N's whole-mesh input paradigm. Although their the ICML version's Section 5.6 mentions patch-based processing, two crucial differences exist: (1) Their approach resembles mesh partitioning rather than our fine-grained node-level processing, which works better for complex geometries where partition may vary significantly; (2) Their patch-based method was actually introduced during the review process (see their rebuttal  on OpenReview, which is visible to everyone only after June 18, 2025 ) and was absent from their February 2025 version. This timeline strongly  confirms the originality of our approach.
>
> Finally, we would like to highlight additional contributions of our work, including: (1) a fully label-free training approach that relies solely on monitor function values, in contrast to methods requiring FEM gradient computations coupled with Firedrake; (2) a mesh-smoothing-inspired iterative optimization scheme that demonstrates consistent improvement across a variety of problem settings; and (3) zero-shot generalization capability, with our model trained on a limited dataset achieving effective performance across multiple PDE cases, compared to approaches that often require case-specific training.
>
> We hope these explanations will help reassess our work's novelty and originality. Below we address your other constructive comments point-by-point (marked as **Q** for questions).
>
> ### **Response to Q1**
>
> We appreciate the clarification request. As described in Section 3.2 (lines 200–205), our training data consists of pre-generated local node patches extracted from high-quality reference meshes (see Figure 10). During training, these patches are treated as static samples—there is no dynamic mesh update or policy rollout. Therefore, the training distribution remains stable, and there is no distribution shift between epochs.
>
> ### **Response to Q2**
>
> In fact, in Figure 4 we have presented the time efficiency of different mesh adaptation methods across various mesh sizes, where our method's timing already includes interpolation operations. The results in Figure 4 demonstrate that our interpolation process does not introduce significant computational overhead, enabling our method to maintain highly efficient single-iteration performance while also achieving substantial reductions in multi-iteration adaptation time.
>
> To provide further evidence, we have conducted timing experiments across different mesh sizes. The results, presented below, show that the main bottleneck is I/O between CPU and GPU, while both inference and interpolation times remain low. These findings confirm the scalability of our method, and we plan to further optimize I/O bottlenecks in future work.
>
> | Element number | Infer. time (mean±std ms) | Inter. time (mean±std ms) | IO. time (mean±std ms) |
> | -------------- | ------------------------- | ------------------------- | ---------------------- |
> | 1478           | 14.3±0.119                | 0.748±0.0154              | 48.7±0.839             |
> | 2396           | 17.9±0.116                | 1.30±0.301                | 75.9±0.853             |
> | 4126           | 25.5±0.448                | 2.49±0.0211               | 136±0.387              |
> | 9248           | 43.9±0.669                | 7.17±0.0703               | 302±0.832              |
> | 36098          | 354±1.88                  | 60.0±0.603                | 1010±1.55              |
>
> ### **Response to Q3**
>
> Thank you for your valuable feedback. Although we have not theoretically analyzed how to guarantee error reduction without mesh tangling, we have demonstrated the excellent performance of our model in error reduction and avoiding mesh tangling through extensive and diverse experiments. Specifically, we have verified the model's performance under varying flow field solutions, mesh resolutions, and mesh shapes, all of which confirm that our method achieves improved solution accuracy without any observed mesh tangling. Overall, our proposed method exhibits stronger robustness and generalization compared to existing traditional methods and ML-based mesh adaptation approaches.
>
> To further illustrate the generalization capability of our model across different scenarios, we have added the following new experiments:
>
> 1. **Random Geometry Test**: On 1,000 randomly generated meshes (with 3–6 edge polygons and complex flow fields), our method improved solution accuracy in 807 cases — the highest among all baselines.
> 2. **Real-world Case**: On a challenging tsunami simulation case with complex coastal boundaries, our model successfully adapted the mesh without producing any invalid elements.
> 3. **Non-convex Test**: On 100 non-convex domains prone to mesh entanglement, our method achieved the best error reduction performance and produced no tangled meshes.
>
> These results indicate that our model offers not only strong generalization but also practical robustness in avoiding mesh tangling under a wide range of conditions.
>
> ------
>
> We are truly grateful for your detailed and thoughtful comments. They have given us a clearer perspective on both the strengths and limitations of our work. We look forward to your follow-up and are happy to provide further clarification if needed.

---

> > ### Comment · Reviewer_dnRz · 2025-08-05
> >
> > Dear authors, thank you for clarifying the timeline, originality and differences of your work in response to my pointing out potential overlaps in my review
> > Thank you for the clarification on the training procedure and pointing out figure 4 I had missed it, the model time is comparable to U/M2N.
> > I update my score to be less negative outlier, unfortunately I can't go higher as the novelty is still incremental compared to the UM2N baseline and existing SOTA methods.

---

> > > ### Author Response · Authors · 2025-08-08
> > >
> > > Dear Reviewer dnRz,
> > >
> > > I hope this message find you well. In our response on August 6, we elaborated on the **novelty** relative to UM2N and state-of-the-art methods, detailed the unsupervised M-Uniform loss, the novel node-patch framework, and the dynamic termination scheme, and highlighted the zero-shot generalization results.
> > >
> > > As the final evaluation approaches, we would like to confirm whether our additional explanations have been fully taken into account, and we are happy to discuss any remaining concerns or points that may need further clarification.

---

### Official Review · Reviewer_AXEC · 2025-07-06

**Clarity:** 2
**Significance:** 3
**Originality:** 3
**Rating:** 4
**Confidence:** 5

**Summary:**

In this paper, the authors propose an unsupervised and generalizable mesh movement network named UGM2N. The method utilizes a patch-based approach with an M-Uniform loss, which encodes local equidistribution properties. The proposed method is validated on various PDEs with different geometries and resolutions. The experiment results show that the UGM2N demonstrates generalizability and robustness abilities.

**Questions:**

- In the mesh reconstruction process, are there any discontinuities around the patch boundaries? How the proposed method handle these potential issues?
- Though the equal-distributed theory should make the mesh movement tangling free if the constraints are strictly satisfied, however, it is difficult for a learning-based method to achieve. In practice, how does the proposed method handle the mesh tangling issue? It will help to better understand the boundary of the proposed method if the authors can provide some tangled cases as well (e.g., similar to the tangling test in UM2N section 5.3)
- How does the proposed method perform apply to real-world cases with arbitrary geometry boundaries? (e.g., similar to tsunami simulation cases in UM2N)
- For the Poisson cases, the error reduction rates have a large variance between different u_exact,  what do the authors think induces this large variance?
- For the time-dependent cases, does the proposed method always start to move a mesh from its uniform version, or the moved mesh is fed into the model autoregressive to next timestep?
- For the moving cylinder and supersonic flow cases, how do the benchmark methods (MA, M2N, UM2N) perform?
- Regarding the iterative mesh adaptation, how is the efficiency of the multiple iterations (including the cost of model inference and Delaunay Triangulation-based interpolation )? And just wondering why does the error reduction show a non-monotonic trend i.e., not a gradual convergence trend?

**Ethical Concerns:**

["NO or VERY MINOR ethics concerns only"]

**Final Justification:**

The authors’ rebuttal addressed my concerns. I would like to raise my score to borderline accept.

**Limitations:**

yes

**Quality:**

3

**Strengths And Weaknesses:**

- Strengths

The patch-based idea and unsupervised M-Uniform loss are intuitive, and the results verify that they are effective in training a learning-based mesh movement model in an unsupervised manner.

- Weaknesses

There are some missing evaluations. The proposed method is mainly applied to simple regular shape geometries such as circles and squares. It is not clear how the proposed method performs when applied to arbitrary boundaries in real-world cases. There is no discussion about how the proposed method handles mesh tangling or what the skill boundary of the UGM2N handling mesh tangling. The large variance of error reduction of the experiments is another concern. Please see more detailed comments in the question section. Would be happy to improve the score if the under-evaluation concerns are addressed.

---

> ### Author Rebuttal · Authors · 2025-07-31
>
> Thank you for your  constructive comments on our paper. We sincerely appreciate the  effort you have dedicated to reviewing our work. Below, we address each of your questions and concerns in detail, with clarifications, new experiments, and implementation insights, where **Q** represents the question and **W** represents the weakness. We hope that our clarifications and additional experiments may encourage you to reconsider your score.
>
> ### **Response to Q1**
>
> We apologize for the unclear description of the reconstruction process. According to Fig. 1, our method appears to reposition all mesh nodes (including boundary nodes) of each patch back onto the mesh; however, as stated in *Line 165* and *Algorithm 1*, we actually only reposition the central nodes, thereby avoiding issues arising from patch boundary conflicts.
>
> ### **Response to Q2**
>
> We greatly appreciate your constructive feedback. First, we would like to address the issue of mesh tangling. Our model utilizes an iterative mesh adaptation process; however, we observed that excessive iteration counts lead to progressive instability and the generation of tangled elements. To resolve this issue, we have implemented a monitoring mechanism for $\mathcal{L}_{\mathrm{var}}\left(\mathcal{M}^{\prime}\right)$ on the mesh and adopted dynamic termination to prematurely halt the adaptation process.
>
> Second,  we have conducted the tangle test you recommended in Section 5.3 of UM2N for further validation. We generated 100 meshes, each containing 8 nodes, with four located at the corners of a unit square and the remaining four randomly sampled within the unit square. The flow field was constructed using a mixture of Gaussian distributions. The results are presented below, demonstrating that both our model and UM2N consistently produced valid meshes. In contrast, the MA method struggles with non-convex cases and only generates valid meshes in limited scenarios.
>
> |                           | MA            | UM2N  | UGM2N |
> | - | - | - | - |
> | Tangling ratio per mesh (mean±std) | 6.63%±10.91% | **0%** | **0%** |
> | Valid mesh | 41 | **100** | **100** |
>
> ### **Response to Q3**
>
> This is an extremely constructive suggestion. We have added the Tohoku tsunami simulation case adopted in UM2N. The results show that:
>
> 1. Our model accurately captures wave propagation, with mesh element sizes dynamically adapting to the wave movement;
> 2. In this complex boundary simulation, our model exhibits no mesh tangling, ensuring mesh validity throughout;
> 3. The adapted meshes generated show no significant element distortion, further guaranteeing mesh quality.
>
> We also intended to include Columbia River and North Sea cases, but their bathymetry datasets were temporarily inaccessible. In the revised version, we will make every effort to include these cases and present their results in the supplementary materials.
>
> ### **Response to Q4**
>
> Thank you for your comments and feedback.
>
> First, we analyzed the reason for the high variance in the results on the Poisson equation. We have found that this discrepancy primarily stems from the fact that we did not update the mesh boundaries, while the distribution of those solutions leads to large Hessian values at the mesh boundaries. To illustrate this point, we allowed mesh nodes to move tangentially along the boundaries and retested the model's performance. The new experimental results show that when further permitting bound node movement, the model's performance improved significantly.
>
> | PDE    | Poisson1 | Poisson2 | Poisson3 | Poisson4 | Poisson5 | Poisson6 | Poisson7 |
> | - | - | - | - | - | - | - | - |
> | ER (%) | 17.49    | 14.82    | 13.08    | 15.98    | 11.18    | 17.85    | 15.69    |
>
> Secondly, we would like to point out that the r-adaptive mesh method struggles to maintain a consistent error reduction ratio across different problems due to significant variations in physical properties, error distributions, and length scales among different flow fields. Its effectiveness heavily depends on the local characteristics of the flow field and the accuracy of the error indicator. Moreover, constrained by the requirement to preserve mesh topology, it cannot adapt to more complex error structures by adjusting the number of mesh elements. **While the r-adaptation is effective, it faces challenges in ensuring consistent error reduction across varying flow fields.**
>
> Beyond highlighting this issue, we believe a more critical question is whether the method can reliably reduce errors (e.g., some approaches may degrade accuracy in certain scenarios), as this reflects the algorithm's robustness. To evaluate this, we tested the adaptation performance on 1,000 randomized cases (with random boundaries and flow fields). The results show that our method achieved effective accuracy improvement in 807 cases, demonstrating its robustness across diverse scenarios.
>
> ### **Response to Q5**
>
> The adapted mesh is computed from **a uniform mesh** at each step (the same as M2N and UM2N). At every time step, we project or interpolate the flow field from the previous time step onto the uniform mesh, then compute the current time step's mesh and use it as the input for the next step.
>
> An autoregressive approach is promising, as it could avoid an interpolation step. However, since the input mesh would become more diverse, the model would need to handle a wider variety of mesh distributions—making it challenging to construct a dataset that covers sufficient cases. This issue is certainly worth further investigation in future research.
>
> ### **Response to Q6**
>
> Thank you for your suggestion. We have evaluated the performance of the benchmark methods on these two  cases, with the following results:
>
> 1. MA: For both  cases, the MA method exhibited divergence behavior during the MA equation solving and failed to generate valid adapted meshes.
> 2. M2N: In the moving cylinder case, the M2N did not demonstrate consistent behavior. While it could generate well-adapted meshes at certain time steps, it produced highly distorted mesh elements at others. For the supersonic case, the M2N could capture the shock position, but with lower mesh node density at the shock compared to UGM2N.
> 3. UM2N: In the supersonic case, although the UM2N could identify the shock position, it introduced  element distortion in irrelevant regions. For the moving cylinder case, while it could fit the flow field distribution, excessive adjustments resulted in distorted mesh elements.
>
> These comparisons further demonstrate the robustness of UGM2N under diverse flow conditions and complex boundaries. Additional results will be included in the revised supplementary materials.
>
> ### **Response to Q7**
>
> **Efficiency Analysis**
>
> We appreciate your reminder regarding the insufficient discussion on time efficiency. In Fig. 4, we present the mesh adaptation time (multiple epochs) of UGM2N under various mesh sizes, where the dashed line indicates the time for a single epoch. It can be observed that the single-epoch adaptation efficiency of our model is higher than that of the M2N and UM2N, allowing us to maintain good efficiency even under multi-epoch adaptation.
>
> To further analyze the time allocation of different operations within a single round, we have constructed meshes with varying numbers of mesh elements to test model inference time (as "infer. time"), mesh node interpolation time (as "inter. time"), and I/O time, and the results are shown below:
>
> | Element number | Infer. time (ms) | Inter. time (ms) | IO. time (ms) |
> | - | - | - | - |
> | 1478           | 14.3±0.12       | 0.748±0.015     | 48.7±0.84    |
> | 2396           | 17.9±0.12       | 1.30±0.30      | 75.9±0.85    |
> | 4126           | 25.5±0.45       | 2.49±0.021      | 136±0.39     |
> | 9248           | 43.9±0.67       | 7.17±0.070      | 302±0.83     |
> | 36098          | 354±1.88         | 60.0±0.60       | 1010±1.55     |
>
> It can be observed that, in a single adaptation epoch, the majority of the time is consumed by I/O operations (data transfer between CPU and GPU), while model inference time and interpolation time remain relatively low. The I/O time consists of two operations:
>
> 1. Transferring mesh data (typically stored on the CPU) to the GPU before adaptive adaptation begins.
> 2. Transferring the newly adapted mesh nodes from the GPU back to the CPU for interpolation.
>
> If a GPU-based Delaunay triangulation interpolation function were available, the second data transfer could be avoided, significantly improving our model's inference efficiency. However, we have not found a mature GPU implementation of such a function, and developing one falls outside the scope of this work. In the future, we will focus on developing this functionality to further enhance the efficiency of our method.
>
>
> **Convergence**
>
> This is an insightful and important observation.  We suggest that this phenomenon could potentially relate to data shifts occurring during adaptation. Our model was trained on uniform meshes, and  the distribution of adapted  mesh elements progressively deviates with increasing iterations. This distributional shift introduces **untrained mesh node distributions**, causing randomized model outputs that may subsequently lead to mesh distortion and eventual tangling.
>
> The most straightforward  solution  is our dynamic termination strategy. This strategy automatically halts the adaptation process when the  $\mathcal{L}_{\mathrm{var}}\left(\mathcal{M}^{\prime}\right)$ ceases to decrease, thereby preventing further quality degradation. This solution effectively addresses the convergence challenges while maintaining computational efficiency.
>
> ------
>
> Thank you again for your careful review and valuable feedback. We hope our responses have addressed your concerns, and we welcome any additional questions or suggestions you may have.

---

> > ### Comment · Reviewer_AXEC · 2025-08-05
> > **Response**
> >
> > Thanks for the authors' detailed reply, which addresses my concerns. I will raise my score to borderline accept.

---

### Official Review · Reviewer_Htea · 2025-07-06

**Clarity:** 3
**Significance:** 2
**Originality:** 3
**Rating:** 4
**Confidence:** 3

**Summary:**

This paper introduces UGM2N, an unsupervised mesh movement network for adaptive mesh generation in PDE solvers. It removes the need for supervised data by introducing a localized node-patch-based learning framework and a novel M-Uniform loss that enforces local mesh equidistribution. The method shows strong generalization across PDEs, mesh geometries, and resolutions, outperforming prior supervised methods (M2N, UM2N) and traditional Monge-Ampère-based techniques in many cases.

**Questions:**

Provide stronger theoretical analysis on convergence or optimality.

Evaluate performance on highly irregular or anisotropic meshes to assess robustness.

**Ethical Concerns:**

["NO or VERY MINOR ethics concerns only"]

**Final Justification:**

great rebuttal and thanks for the efforts on the improvement; i have raised my score.

**Limitations:**

yes

**Quality:**

2

**Strengths And Weaknesses:**

Strengths
Novel unsupervised formulation: The M-Uniform loss and node-patch approach are original and practically motivated.

Strong empirical results: Demonstrates excellent generalization across a wide range of PDEs and mesh configurations.

Efficiency: The method remains computationally efficient despite its iterative adaptation scheme.

Well-structured experiments: Includes diverse cases, ablation studies, and rigorous baseline comparisons.

Weaknesses
Limited theoretical depth: While the M-Uniform loss is well motivated, the theoretical analysis remains relatively basic (mostly variance decomposition) and lacks deeper convergence or approximation guarantees.

Boundary limitation: The method currently ignores boundary nodes, which limits applicability to many real-world problems where boundary behavior is critical.

Assumption on mesh regularity: The approach assumes quasi-uniform meshes where node degrees are roughly constant—this assumption may fail for highly anisotropic or unstructured meshes, but this limitation is not sufficiently addressed in practice.

Architecture simplicity: The graph transformer model used is relatively basic; it is unclear how much the performance comes from the loss function versus the model itself.

---

> ### Author Rebuttal · Authors · 2025-07-31
>
> The authors gratefully acknowledge the reviewers' valuable feedback, which helped us refine and strengthen the paper. Below we provide our responses to the reviewers' comments, where questions are labeled as **Q** and weaknesses are denoted as **W**.
>
> ### **Response to W1\Q1**
>
> We sincerely appreciate the reviewer’s valuable suggestions regarding convergence and approximation guarantees. We fully acknowledge the concerns and recognize that Appendix A.2 currently provides only a preliminary theoretical analysis. As noted in our paper, our method draws inspiration from conventional mesh smoothing techniques, which also optimize mesh quality by minimizing a scalar function defined over local elements. However, our approach differs fundamentally in that we optimize the monitor function integral approximation $\mathcal{L}\_{K} =m_{K}|K|$  (Eq. 6), whereas traditional smoothing focuses on geometric quality metrics (e.g., aspect ratio, minimum angle).
>
> While this similarity might suggest that convergence or approximation guarantees could be readily derived, such analysis proves challenging in practice. Under a serial implementation—where nodes are updated sequentially—each iteration provably improves $\mathcal{L}\_{\mathrm{var}} ( \mathcal{P}\_{i} )$ within a local neighborhood, theoretically ensuring convergence. However, the final approximation accuracy is inherently limited by the model’s training error, as the learned mapping cannot guarantee exact minimization (i.e., $\mathcal{L}\_{\mathrm{var}} ( \mathcal{P}\_{i} )=0$) for arbitrary node patches. Moreover, this serial approach is computationally inefficient due to its non-vectorizable nature.
>
> In contrast, our practical implementation employs Jacobi-style parallel updates, which significantly improve efficiency but introduce inherent analytical difficulties. Although the model effectively optimizes local node distributions, the convergence behavior and approximation error remain challenging to characterize theoretically. Unlike Laplacian smoothing, which permits linear analysis, our transformation $T$ operates as a black-box nonlinear function, making it challenging to verify whether it possesses certain properties (e.g., whether it constitutes a contraction mapping).
>
> Further complicating the analysis are the practical challenges of parallel updates, including data races (concurrent modifications of shared regions), update conflicts (local improvements causing global degradation), and information delays (reliance on stale neighbor positions). Additionally, while local node neighborhoods are often geometrically convex, the Hessian of our objective function may not be, precluding standard convex optimization arguments.
>
> These challenges reflect broader theoretical difficulties inherent to adaptive mesh optimization, and are not unique to our model. Nevertheless, as demonstrated empirically in Section 5, our method achieves robust convergence and superior adaptation quality in practice. We have expanded Appendix A.2 to further clarify these theoretical considerations in response to the reviewer’s insightful feedback.
>
> Despite the aforementioned theoretical challenges, we have empirically evaluated the convergence behavior of our model by monitoring the evolution of the nodal displacement norm  $\delta_k=||\mathbf{X}^{k+1}-\mathbf{X}^{k}||$ across iterations, where $\mathbf{X}^{k}$ denotes the mesh node positions at optimization step $k$. As demonstrated in the results for the Helmholtz equations (shown below), the consistent decay of $||\mathbf{X}^{k+1}-\mathbf{X}^{k}||$ toward zero with increasing $k$ provides empirical evidence of algorithmic convergence. However, when $k$ increases (greater than 10), we also observed the phenomenon that $\delta_k$ fails to converge or even increases in test cases. In such scenarios, our proposed dynamic termination method can effectively address this issue, thereby enhancing the model's convergence.
>
> | PDE        | $\delta_0$ | $\delta_1$ | $\delta_2$ | $\delta_3$ | $\delta_4$ | $\delta_5$ | $\delta_6$ | $\delta_7$ | $\delta_8$ | $\delta_9$ |
> | - | - | - | - | - | - | - | - | - | - | - |
> | **Helmholtz1** | 0.121      | 0.089      | 0.070      | 0.060      | 0.053      | 0.047      | 0.045      | 0.042      | 0.045      | 0.047      |
> | **Helmholtz2** | 0.110      | 0.081      | 0.068      | 0.058      | 0.051      | 0.046      | 0.042      | 0.039      | 0.036      | 0.034      |
> | **Helmholtz3** | 0.105      | 0.075      | 0.058      | 0.047      | 0.038      | 0.032      | 0.028      | 0.026      | 0.024      | 0.022      |
> | **Helmholtz4** | 0.120      | 0.062      | 0.041      | 0.032      | 0.028      | 0.025      | 0.024      | 0.022      | 0.022      | 0.022      |
> | **Helmholtz5** | 0.122      | 0.060      | 0.038      | 0.029      | 0.025      | 0.023      | 0.023      | 0.023      | 0.023      | 0.023      |
>
> ### **Response to W2**
>
> We sincerely appreciate your valuable feedback. When dealing with complex boundaries, ensuring the geometric consistency of mesh boundaries is a challenge. The fundamental difficulty stems from the inevitable prediction errors in ML models, where even minor deviations of predicted boundary nodes can significantly compromise accuracy by displacing boundary layers - the critical thin regions near surfaces where viscous, thermal or electromagnetic effects dominate. This justifies our design choice to avoid direct boundary node prediction.
>
> Our new experimental validation reveals that while simple clip operations (projecting nodes onto boundaries) show minimal impact (list below) for internally-dominated flows like Helmholtz and Burgers cases in our paper, they provide substantial improvements for cases involving rapidly varying boundary conditions (detailed in Response Q3 to Reviewer mhZx). These findings indicate that, while our current treatment provides reasonable robustness, there remains ample opportunity to further improve boundary prediction through learned or hybrid approaches.
>
> | PDE        | Burgers0 | Burgers1 | Helmholtz1 | Helmholtz2 | Helmholtz3 | Helmholtz4 | Helmholtz5 |
> |-|-|-|-|-|-|-|-|
> | **ER w.o. Clip (%)** | 32.22 | 30.19 | 14.11 | 13.15 | 15.03 | 14.09 | 16.98 |
> | **ER with Clip (%)** | 35.27    | 30.18    | 16.68      | 16.17      | 15.94      | 15.57      | 17.34      |
>
> ### **Response to W3\Q2**
>
> Thank you for your feedback. Following your suggestions, we have added two new experiments to evaluate the performance of our method on both highly irregular and anisotropic meshes. The detailed experimental setup and results are presented below.
>
> **Highly irregular mesh**
>
> We uniformly sampled mesh points within a unit rectangular domain and generated meshes via Delaunay triangulation, followed by mesh smoothing to optimize mesh quality. The meshes generated this way contain numerous mesh points with different degrees, making them fail to satisfy mesh regularity.
>
> We generated 10 samples in total, with the flow field solution on each sample given by $\cos(2  \pi  (x - \mu_x))$, where $\mu_x$ is a random value in [0, 1]. The error reduction results of our model are shown below:
>
> |            | Case 1 |   Case 2   | Case 3 |  Case 4   | Case 5 |  Case 6  |    Case 7 |    Case 8 |  Case 9   |   Case 10  |
> | ---------- | ------ | ---- | ------ | ---- | ------ | ---- | ---- | ---- | ---- | ---- |
> | **ER (%)** | -2.07 | 16.07 | 10.69 | 11.11 | 18.10 | 14.24 | -2.10 | 8.46 | 9.84 | 7.16 |
>
> The results demonstrate that, although our model was trained on regular meshes, it can generalize well to highly irregular meshes, achieving effective error reduction in 8 out of 10 cases.
>
> **Anisotropic meshes**
>
> We have added a new test case for anisotropic meshes, where the flow field solution is given by $u(x,y)=(1-e^{\frac{-(x-0.5)^2}{0.01}})((x-0.5)^2-1), [x,y] \in [0,1] \times [0,1] $.  First, we generate preliminary stretched mesh elements through anisotropic adaptation to create anisotropic meshes. These generated anisotropic meshes are then fed into our model to further optimize the node distribution to match the characteristics of the solution. In this solution, the Hessian values reach their maxima at approximately x = 0.3, 0.5, and 0.7. Our UGM2N model can capture these three locations more accurately compared to the initial anisotropic meshes. After solving with the optimized anisotropic meshes, the error reduction (absolute value) for each model is shown in the table below, demonstrating that our model has the potential to be applied to anisotropic meshes.
>
> |                 | MA   | M2N  | UM2N | UGM2N    |
> | --------------- | ---- | ---- | ---- | -------- |
> | Error reduction | Fail |Fail      | 0.02 | **0.09** |
>
> ### **Response to W4**
>
> We sincerely appreciate your valuable feedback. In fact, our ablation studies in Section 4.4 have already evaluated the impact of different loss functions on model performance under identical architectures. The results demonstrate that our proposed loss function outperforms those used in previous methods when maintaining the same network structure. We would be happy to conduct further comparisons of loss functions and architectures (including M2N and UM2N) to validate our approach, if of interest.
>
> Furthermore, compared to prior approaches, our model architecture is significantly simpler. This design choice enables highly efficient inference per adaptation epoch (as shown in Figure 4), while maintaining substantially better computational efficiency than MA methods even for multiple adaptation rounds.
>
> ------
>
> We greatly appreciate the time and effort the reviewers have devoted to evaluating our work. Your constructive feedback has deepened our understanding of the key issues and has been extremely helpful. We remain open to further suggestions and look forward to your continued guidance.

---

> > ### Comment · Reviewer_Htea · 2025-08-01
> > **great rebuttal**
> >
> > great rebuttal and thanks for the efforts on the improvement; i have raised my score.

---

> > > ### Author Response · Authors · 2025-08-01
> > >
> > > Thank you for recognizing our efforts. Your valuable comments have guided us in significantly improving the quality of the paper. We will continue to refine the manuscript in the revised version. Once again, thank you for your time and effort during the review process. ( ﹡ˆoˆ﹡ )

---

### Official Review · Reviewer_mhZx · 2025-07-13

**Clarity:** 2
**Significance:** 2
**Originality:** 3
**Rating:** 4
**Confidence:** 4

**Summary:**

This paper presents UGM2N, an unsupervised neural network for mesh movement in numerical simulations. The method introduces a "node patch" representation to process local mesh neighborhoods independently. It proposes a physics-constrained loss function, the M-Uniform loss, which enforces mesh equidistribution without requiring pre-adapted reference meshes for supervision. The goal is to create a generalizable framework that works across different partial differential equations (PDEs) and mesh geometries, improving simulation accuracy and efficiency.

**Questions:**

1. The M-Uniform Loss has a scaling hyperparameter λ set to 100. How was this value chosen, and how sensitive is the model performance to this value? An ablation study would be necessary to understand the robustness here.
2. The M-Uniform loss calculation uses the Hessian matrix of the flow field solution u. This seems to require that the PDE expression is known. How can the method generalize to systems where the governing PDE is not known explicitly? If it cannot, I seriously question the generalizability claimed.
3. In Table 1, the performance is sometimes weaker than baselines. For instance, in the first Burgers' equation case, M2N achieves 44.82% ER while your method gets 32.22%. Can you give some intuition for why the model does not perform as well in these cases? A clear explanation of failure modes would make me reconsider my score.
4. The paper evaluates on complex geometries like cylinder flow and airfoil, which is good. However, the training is done only on four simple analytical flow fields. Is it possible the model overfits to certain features of these simple training functions? Seeing generalization from more complex training data would strengthen the paper.

**Ethical Concerns:**

["NO or VERY MINOR ethics concerns only"]

**Final Justification:**

The reviewers have addressed my concerns in their rebuttal, and they have promised to improve the writing for the final version.

**Limitations:**

yes

**Quality:**

2

**Strengths And Weaknesses:**

Strengths:
- The introduction of "node patches" is a novel approach for mesh adaptation. It processes the mesh locally, which simplifies the learning problem and is well-suited for parallel computation.
- The method is unsupervised, learning from initial meshes and flow fields directly. This removes the dependency on generating high-quality adapted meshes with traditional, costly methods, which is a major practical advantage.
- The experiments show the model's performance on varying mesh geometries, including different shapes (airfoil, cylinder) and resolutions, which is a good demonstration of its potential geometric flexibility.

Weaknesses:
- The paper has some presentation issues. Many specialized terms from computational physics and deep learning (e.g., "Monge-Ampère based method", "GAT", "Delaunay triangulation") are used without sufficient explanation, which makes the paper difficult to follow for an audience not expert in this specific field. I feel this is paper is not good venue for Neurips, more fitted to a physics journal.
- The performance is not uniformly good. In Table 1, the method is sometimes outperformed by traditional methods (MA) or other learning-based baselines (M2N), which raises questions about the robustness of the proposed approach.
- The experimental analysis feels incomplete. For example, there is no ablation study on key hyperparameters like the scaling constant λ in the proposed M-Uniform loss function.

---

> ### Author Rebuttal · Authors · 2025-07-28
>
> We sincerely appreciate the reviewers' insightful comments and constructive suggestions, which have greatly helped us improve the quality of our paper. Below, we will address each of the weaknesses (**W**) and questions (**Q**) raised in your review comments point by point.
>
> ### **Response to W1**
>
> **Terminology**
>
> We fully acknowledge your concern regarding insufficient explanations of specialized terms. Since our work involves AI for physics, the field encompasses a wide range of specialized terminology due to its interdisciplinary nature, combining concepts from physics, applied mathematics, and computer science. We recognize that additional context would benefit readers with diverse backgrounds. In the revised version, we commit to expand explanations in the main text or appendices  and include targeted references  for further reading.
>
> **Suitability**
>
>  *Machine learning for physics-based simulation* is a thriving subfield within *AI for Science*, as evidenced by numerous seminal works published at NeurIPS (e.g., *SchNet*, *Hamiltonian Neural Networks*, and *Neural ODEs*). The prior AI-based r-adaptation works, *M2N*  and *UM2N*, also appeared in NeurIPS. We believe this work aligns well with NeurIPS’s scope.
>
> ### **Response to W2**
>
> **Robustness**
>
> We acknowledge that our model does not achieve optimal performance in every single case. However, as evaluated from multiple perspectives in the paper, our approach consistently outperforms existing methods across various scenarios, including:
>
> 1. **Flow Field Variations:**
>     - Achieves **9 best results out of 14 cases** (64.3%).
>     - Outperforms existing ML-based methods (*M2N* and *UM2N*) in **12 out of 14 cases** (85.7%).
> 2. **Mesh Scale Adaptability (Fig. 3)**:
>     - Maintains error reduction capability across different flow fields.
>     - Delivers **14 best results out of 20 test cases** (70%).
>     - Surpasses ML-based baselines in **15 cases** (75%).
> 3. **Variable Geometry Tests**: Exceeds both traditional MA methods and ML-based approaches.
>
> These results confirm that **our method offers the most consistent performance overall**, despite minor limitations in isolated cases. We also want to emphasize that achieving perfect performance in *every* scenario is inherently challenging—even traditional methods fail in certain cases (see **Tab. 1**).
>
> **Additional Robustness Validation**
> To further validate robustness, we have conducted a new large-scale test on 1,000 mesh samples, where each sample contains random polygonal shapes (3–6 edges) and Gaussian mixture-based flow fields.
>
> We solved the Helmholtz equation using these flow fields as solutions, with the comparative results shown in the table below. The results show that both MA and UM2N suffer from performance variance across different flow fields and geometries. In contrast, our method demonstrates significantly greater robustness in mesh adaptation.
>
> Notably, our approach achieves the highest Positive ER Ratio (807 successful cases out of 1000), substantially outperforming both he traditional method and UM2N. The M2N results have been excluded from the table due to its inherent limitation in generalizing across arbitrary geometries and flow fields without case-specific training.
>
> | Method           | Error Reduction (mean%±std%) | Positive ER Ratio |
> | - | - | - |
> | MA               | -227.76±468.82 | 0.110       |
> | UM2N             | -50.72±68.36 | 0.245      |
> | **UGM2N (Ours)** | **13.99±21.05** | **0.807** |
>
> ### **Response to W3\Q1**
>
> Thank you for pointing out the limitations in our experimental design. We have conducted an ablation study to systematically evaluate the influence of parameter $\lambda$ on model performance. We define $\lambda$ = 10**scale, with scale values spanning the range from -1 to 3 in increments of 0.5. To ensure  reliability, each $\lambda$ value was tested with 5 repeated experiments. The converged test losses are shown in the table below, which indicate a linear relationship between $\lambda$ and the final converged loss, which aligns with the Eq. (8) of our loss function. This suggests that the value of $\lambda$ has minimal impact on the model's convergence.
>
> | $\lambda$      | 0.1                  | 0.32                 | 1                    | 3.2                  | 10                   | 32                   | 100                  | 320         | 1000        |
> | - | - | - | - | - | - | - | - | - | - |
> | **Test Loss** | (3.89 ± 1.52) × 10⁻⁴ | (1.21 ± 0.22) × 10⁻³ | (4.12 ± 0.98) × 10⁻³ | (1.07 ± 0.08) × 10⁻² | (4.33 ± 2.38) × 10⁻² | (1.73 ± 0.77) × 10⁻¹ | (4.11 ± 1.28) × 10⁻¹ | 1.35 ± 0.47 | 3.20 ± 0.24 |
>
> The results on the  the Poisson and Helmholtz equations are summarized in the below table. Overall, the ER remain relatively stable across different $\lambda$ values, suggesting that our loss function is not highly sensitive to $\lambda$. However, certain cases (e.g., Helmholtz1, Poisson4 and Poisson7) exhibit greater sensitivity to $\lambda$, which we attribute to case-specific effects rather than model limitations, as they prove challenging to optimize across all models.
>
> | PDE    | Helmholtz1   | Helmholtz2   | Helmholtz3   | Helmholtz4   | Helmholtz5   | Poisson1     | Poisson2    | Poisson3     | Poisson4    | Poisson5    | Poisson6    | Poisson7    |
> | - | - | - | - | - | - | - | - | - | - | - | - | - |
> | ER (%) | 13.34 ± 4.48 | 10.89 ± 1.74 | 14.74 ± 0.72 | 13.23 ± 0.63 | 15.53 ± 1.03 | 13.06 ± 2.03 | 8.82 ± 0.68 | 12.17 ± 0.70 | 2.28 ± 1.26 | 9.15 ± 1.36 | 5.40 ± 1.65 | 2.87 ± 1.26 |
>
> ### **Response to Q2**
>
> This is indeed an important point that warrants clarification. In solving PDEs, we typically know the governing  PDE and boundary/initial conditions, while the solution *u* remains unknown. As you correctly noted, for simple problems we may obtain explicit expressions for *u*, but for complex flow problems such explicit solutions are generally unavailable.
>
> However, our method does not require explicit expressions for *u* to compute the Hessian matrix, since we can numerically recover solution derivatives through established numerical differentiation techniques, which is detailed in book *Adaptive Moving Mesh Methods* P266. This numerical method only requires flow field variables at mesh nodes and computes the Hessian (and gradient) values via the least squares fitting method or the Galerkin formulation method, enabling our approach to generalize to arbitrary flow fields.
>
> ### **Response to Q3**
>
> This is an excellent question that provides valuable insights for further model optimization. Our analysis reveals that M2N's superior performance on the first Burgers equation stems from its tendency to generate more deformed meshes (compared to our smoother UGM2N outputs), which better capture abrupt flow variations in this specific case. However, as Fig. 2 demonstrates, this characteristic often leads to mesh distortion in most other cases, ultimately compromising accuracy. This suggests M2N's advantage is case-specific rather than methodological.
>
> Regarding the suboptimal results on Poisson4-7, we identified that our boundary treatment - specifically, not updating boundary nodes - becomes problematic when large Hessian values concentrate near boundaries while interior values remain small. New experimental validation (see table below) shows significant improvement when allowing boundary-constrained node movement. Notably, even without this modification, our model outperforms both M2N and UM2N (which natively permit boundary movement), confirming our approach's fundamental advantages.
>
> | PDE    | Poisson1 | Poisson2 | Poisson3 | Poisson4 | Poisson5 | Poisson6 | Poisson7 |
> | - | - | - | - | - | - | - | - |
> | ER (%) | 17.49    | 14.82    | 13.08    | 15.98    | 11.18    | 17.85    | 15.69    |
>
> Further tests reveal boundary adaptation's differential impact:
>
> - **Helmholtz cases**: Minimal improvement as flow variations predominantly occur in interior regions
> - **Burgers cases**: Substantial ER boost to 35.27% in Case1 but limited gains in Case2
>
> These findings highlight two critical factors: (1) solution characteristics and (2) boundary treatment quality, particularly for curved boundaries where simple assignment proves inadequate and projection becomes necessary. Developing enhanced boundary handling methods for complex geometries emerges as a key direction for future research.
>
>
> ### **Response to Q4**
>
> We appreciate your thoughtful comments. First, while our model was trained on *only 4 flow fields*, the use of node patches as fundamental data units resulted in a substantial dataset of 10,440 samples, which proved sufficient for effective model training. The model's demonstrated generalization capability across multiple test cases, enabled by its few-shot learning ability, eliminates the need to prepare large-scale training datasets.
>
> To further investigate the impact of training data volume on model performance, we have conducted additional experiments with expanded flow field configurations (4, 8, and 16 fields), corresponding to training nodes of [10440, 20880, 41760]. The experimental results demonstrate consistent improvement in our model's generalization capability with increasing training data across Poisson, Helmholtz and Burgers equations, with particularly significant gains observed for Poisson and Helmholtz cases. This inspired us to expand richer datasets in subsequent work to further enhance the model's generalization ability.
>
> | Data number | Poisson | Helmholtz | Burgers |
> | - | - | - | - |
> | 10440       | 5.21    | 9.94      | 30.07   |
> | 20880       | 6.56    | 13.68     | 29.36   |
> | 41760       | **7.43** | **14.21** | **30.56** |
>
> ------
>
> We sincerely thank the reviewers for their thoughtful comments and insightful questions. Your feedback has been invaluable in helping us reflect on and refine various aspects of our work. We look forward to any further discussion or clarification you may require.

---

> > ### Comment · Reviewer_mhZx · 2025-08-01
> > **Thank you**
> >
> > Thank you for your clarification. I have raised my score.

---

> ### Author Response · Authors · 2025-08-01
>
> We are very pleased that we have addressed your concerns and achieved an improved score. Your comments have greatly enhanced the quality of our paper. We will include the additional content in the revised version of the manuscript. Finally, we would like to thank you once again for your efforts during the review process. (＾▽＾)

---

### Note · Authors · 2025-08-12

Response to  Reviewer **YRmJ**'s comment (09 Aug 2025, 18:12):

As the discussion time was limited, we wish to offer a brief clarification that may help better frame our earlier response.

In the **Questions** section of the original review,  reviewer YRmJ asked about the **cause** of why the mathematically symmetric solutions cos(2πy) and cos(2πx) produced different results:

> ... Could the authors elaborate on **the potential cause** of this discrepancy? Is this slight performance difference **a result of** inherent asymmetries in the training dataset, **the underlying mesh triangulation**, or another artifact of the learning process?

In our rebuttal, we indicated that the cause is **inherent asymmetries in the underlying mesh triangulation**—since a 90° rotation of the test mesh cannot achieve perfect overlap with the original. When solving PDEs, it is well established that the mesh can directly influence solution accuracy; once the mesh changes, **identical accuracy is not guaranteed**, particularly when mesh adaptation is applied.

We understand that reviewer YRmJ later noted our explanation did not “resolve” the issue. We would like to clarify that the observed difference arises from the mesh itself, rather than from limitations of the proposed model. While we additionally explored rotational invariance as a related phenomenon—finding that AI-based methods are indeed less rotation-invariant than traditional methods—this was intended as supplementary insight rather than a direct solution, as the original question primarily sought a cause.

Finally, we appreciate YRmJ’s attention to rotational invariance, which is certainly an interesting direction. In our work, however, the main focus is on **unsupervised learning and generalization**, and we believe these contributions are thoroughly supported by both the paper and the rebuttal, demonstrating the efficiency and robustness of our model across steady and unsteady flow fields, as well as varying flow conditions and meshes.

---

### Decision · Program_Chairs · 2025-09-17

**Decision:**

Accept (poster)

**Comment:**

The authors proposed an unsupervised mesh adaptation approach for numerical simulations. To process the whole mesh the method considers local mesh neighnorhoods. To train the model for mesh adaptation the approach uses a physics-constrained loss function to enforce mesh equidistribution. The authors demonstrated that the method works across different PDEs and mesh geometries and can significantly improve accuracy of simulations.

As a strengths of the paper I would highlight
- the paper is rather clearly written and properly motivated
- the method is unsupervised, so as a result the method does not require to generate high-quality adapted mesh
- the experimental section convincingly demonstrates that the proposed approach outperforms other correcting schemes
-  the model demonstrates generalization capability across multiple test cases, enabled by its few-shot learning ability, so it eliminates the need to prepare large-scale training datasets

As a weaknesses I would highlight
- the text should be further polished. E.g. there are many specialised terms in the text that could be not so familiar for DL/ML community
- there is no theoretical analysis confirming convergence or approximation guarantees
- the method currently does not have specific mechanisms to work with boundary nodes, which could be crucial for some real-world problems

The main reasons for the accept decision are
- unsupervised approach for mesh adaptation with good generalization capabilities across multiple test cases
- the approach is rather computationally efficient
- extensive empirical evaluation

During the discussion with the reviewers several issues were raised that should be addressed in the final version of the paper. Here are some issues
- reviewer mhZx proposed to include explanations of some specific terms from computationsl physics such as GAT, Monge-Ampère based method, etc.
- reviewer Htea proposed to consider boundary limitation issues of the proposed method
- the authors did a lot of additional experiments in response to authors questions. These results should be included in the main text and in the appendix